



# Retrievals of Tropospheric Ozone Profiles from the Synergic Observation of AIRS and OMI: Methodology and Validation

Dejian Fu[1], Susan S. Kulawik[2], Kazuyuki Miyazaki[3], Kevin W. Bowman[1], John R. Worden[1], Annmarie Eldering[1], Nathaniel J. Livesey[1], Joao Teixeira[1], Fredrick W. Irion[1], Robert L. Herman[1], Gregory B. Osterman[1], Xiong Liu[4], Pieternel F. Levelt[5,6], Anne M. Thompson[7], Ming Luo[1]

[1]NASA Jet Propulsion Laboratory, California Institute of Technology, Pasadena, California, USA
[2]Bay Area Environmental Research Institute/NASA Ames Research Center, Mountain View, California, USA
[3]Japan Agency for Marine-Earth Science and Technology, Yokohama, Japan
[4]Harvard-Smithsonian Center for Astrophysics, Cambridge, Massachusetts, USA
[5]Royal Netherlands Meteorological Institute, De Bilt, 3731 GA, The Netherlands
[6]Faculty of Civil Engineering and Geosciences, University of Technology Delft, Delft, 2628 CN, The Netherlands
[7]NASA Goddard Space Flight Center, Greenbelt, Maryland, USA

*Correspondence to*: Dejian Fu (dejian.fu@jpl.nasa.gov)

**Abstract.** The Tropospheric Emission Spectrometer (TES) on the A-Train Aura satellite was designed to profile tropospheric ozone and its precursors, taking measurements from 2004 to 2018. Starting in 2008, TES global sampling of tropospheric ozone was gradually reduced in latitude with global coverage stopping in 2011. To extend the record of TES, this work presents a multispectral approach that will provide $O_3$ data products with vertical resolution and measurement uncertainty similar to TES by combining the single-footprint thermal infrared (TIR) hyperspectral radiances from the Aqua Atmospheric Infrared Sounder (AIRS) instrument and the ultraviolet (UV) channels from the Aura Ozone Monitoring Instrument (OMI). The joint AIR+OMI $O_3$ retrievals are processed through the MUlti-SpEctra, MUlti-SpEcies, MUlti-SEnsors (MUSES) retrieval algorithm. Comparisons of collocated joint AIRS+OMI and TES to ozonesonde measurements show that both systems have similar errors, with mean and standard deviation of the differences well within the estimated measurement uncertainty. AIRS+OMI and TES have slightly different biases (within 5 parts per billion) versus the sondes. Both AIRS and OMI have wide swath widths (~1,650 km for AIRS; ~2,600 km for OMI) across satellite ground tracks. Consequently, the joint AIRS+OMI measurements have the potential to maintain TES vertical sensitivity while increasing coverage by two orders of magnitude, thus providing an unprecedented new dataset to quantify the evolution of tropospheric ozone.



# 1 Introduction

Long-term records of the vertical distribution of ozone are essential for quantifying the impact of changes in tropospheric ozone on air quality and climate, driven recently by rapid industrialization in Asia concurrent with reductions in ozone precursor emissions in North America and Europe (Jacob et al., 1999; Wild and Akimoto, 2001; Akimoto 2003; Worden et al., 2008; Fischer et al., 2011; Worden et al., 2011). The A-Train Aura satellite has played an important role in quantifying the atmospheric ozone and advancing our understanding of the processes controlling its distribution. The Dutch–Finnish Ozone Monitoring Instrument (OMI) measures ultraviolet (UV) radiances, which are used to infer a number of species including ozone profiles and columns (Levelt et al., 2006a, b, 2018; Liu et al., 2010 a, b; Huang et al., 2017). These measurements have been used in a number of assimilation systems to constrain both stratospheric and tropospheric ozone distributions (Stajner et al., 2008; Pierce et al., 2009; Huang et al., 2013; Inness et al., 2013; Wargan et al., 2015; Olsen et al., 2016). OMI ozone columns have been used to understand both tropical ozone variability (Chandra et al., 2007; Ziemke et al., 2007) and high-latitude ozone including the unprecedented Arctic ozone loss in 2011 (Manney et al., 2011). The Aura Tropospheric Emission Spectrometer (TES) has a spectral resolution of $0.1$ cm$^{-1}$, the highest infrared spectral resolution among any current nadir sounder, which enables estimation of tropospheric ozone profiles and precursors. TES has advanced a number of Aura science objectives, including detection of tropospheric ozone trends over Asia (Lamsal et al., 2011; Verstraeten et al., 2015), the influences of long-range pollution transport on surface ozone (Parrington et al., 2008, 2009), and the tropospheric ozone response to stratospheric circulation (Neu et al., 2014). The TES record has also played an important role in evaluating chemistry-climate model simulations of present-day ozone distributions and their ozone radiative forcing as part of the Intergovernmental Panel on Climate Change (IPCC) Fifth Assessment Report (AR5; Bowman et al., 2013; Shindell et al., 2013; Young et al., 2013; IPCC, 2014) and in providing constraints on the tropospheric chemistry through data assimilation (Miyazaki et al., 2012; 2014; 2015). TES global observation are limited to a roughly 5-year period (2005–2009) due to instrument aging. TES global sampling of tropospheric ozone was gradually reduced starting in 2008 with global observations ceasing altogether in 2011. Consequently, TES' well-validated global survey record of tropospheric ozone (Worden et al., 2007a; Nassar et al., 2008; Boxe et al., 2010; Verstraeten et al., 2013; Bella et al., 2015) ended in 2011.

The synergy of combining UV and ultra-spectral thermal infrared (TIR) radiances provides an approach to measure to lower tropospheric ozone, a key objective of air quality remote sensing (Worden et al., 2007b; Landgraf and Hasekamp, 2007; Costantino et al., 2017). This capability was demonstrated by Fu et al. (2013) for joint TES+OMI and Cuesta et al. (2013, 2017) for joint Infrared atmospheric sounding interferometer (IASI) and Global Ozone Monitoring Experiment 2 (GOME-2). Ozone profiles from joint TES+OMI retrievals are a part of the standard Earth Observing System (EOS) Aura products from the time period 2005 to 2008, the time period when neither the degradation of TES instrument nor the row anomaly of OMI pixels (Huang et al., 2017; Schenkeveld et al., 2017; Levelt et al., 2018), which provide measurements collocated to TES measurements, played a role.

In this work, we demonstrate that joint Atmospheric Infrared Sounder (AIRS) and OMI retrievals can extend the Aura-EOS TES standard level 2 tropospheric ozone concentration vertical profile products. The retrieved ozone profiles



harnessing the level 1B radiances from AIRS and OMI measurements have vertical resolution and error characteristics similar to the TES instrument on Aura and the prospect of vastly increased spatial coverage.

## 2 TES, AIRS, OMI, and ozonesonde measurements

The NASA A-Train satellites (Aqua, Aura, Cloud-Aerosol Lidar and Infrared Pathfinder Satellite Observation
(CALIPSO), CloudSat, Orbiting Carbon Observatory-2 (OCO-2)) are providing long-term global measurements of the land surface, biosphere, atmosphere, and oceans of the Earth in a near-polar, sun-synchronous, ~700 km altitude orbit whose ascending node has an equator crossing time at around 13:30 p.m. local time. The measurements of three nadir viewing instruments in the A-Train satellites, including Aura-TES, Aura-OMI, and Aqua-AIRS, play essential roles on quantifying atmospheric composition, including $O_3$ and a suite of trace gases to advance understanding of air quality and climate science.

TES is a Fourier transform spectrometer (FTS) that measures the double-sided interferograms of TIR radiances emitted and absorbed by Earth's surface, gases and particles in the atmosphere (Beer et al., 2001). Although TES has both the nadir and limb views, nadir has been the primary scanning geometry used to obtain full vertical and horizontal coverage of Earth's atmosphere. In nadir mode, TES measurements cover four optical filter bands (650–900, 950–1,150, 1,100–1,325, and 1,900–2,250 $cm^{-1}$) with a constant spectral resolution of 0.1 $cm^{-1}$ and a ground pixel size of $5.3 \times 8.5$ $km^2$. The 950–1,150 $cm^{-1}$ spectral

regions include high-density absorption features of the ozone υ3 band (the strongest fundamental band) and minor absorption from interfering species. The υ3 band has been exploited in the tropospheric $O_3$ soundings by a suite of TIR satellite-borne, nadir-viewing instruments including AIRS (Susskind et al., 2003, 2014; Wei et al., 2010), Cross-track Infrared Sounder (CrIS) (Gambacorta et al., 2013), and IASI (Boynard et al., 2009; Dufour et al., 2012; Oetjen et al., 2014; Boynard et al., 2016; Oetjen et al., 2016), as well as the solar occultation satellite-borne (Bernath et al., 2005; Bernath 2017), balloon-borne (Toon 1991; Fu

et al., 2007a), and ground-based (Hannigan et al., 2011) FTSs that quantify the stratospheric ozone layer and the species playing essential role in the stratospheric ozone chemistry (Fu et al. 2007b, 2009, 2011; Sung et al., 2007; Wunch et al., 2007; Allen 2009; Boone 2013; Nassar 2013; Griffin et al., 2017). The spectral resolution of TES (resolving power (RP) 10,500) is significantly higher than the existing TIR and UV space spectrometers including AIRS (RP: 1,200), CrIS (RP: 816), IASI (RP: 5,250), and OMI (RP: 460–803). Benefiting from the Aura afternoon orbit, TES takes measurements around local noon time when the

atmosphere/land thermal contrast is typically higher than other times of the day. As a result, TES has the sensitivity to distinguish between the upper and lower tropospheric $O_3$.

AIRS is a grating spectrometer that measures the Earth's TIR emission in the spectral range of 650–2,665 $cm^{-1}$ (Aumann et al., 2003). It is a cross-track scanning instrument providing measurements with daily global coverage. AIRS atmospheric measurements in the ozone υ3 band provide sensitivity for estimating atmospheric ozone column density. The currently

operational AIRS version 6 retrieval algorithm (Susskind et al., 2003, 2014) estimates the temperature, humidity, and atmospheric composition products using the 45 km resolution, level 2 cloud-cleared radiance products for weather prediction and environmental monitoring. In order to fully exploit the spatial resolution of AIRS measurements, our joint AIRS+OMI ozone



retrievals use single-footprint (i.e., non cloud-cleared) level 1b AIRS infrared radiances with a spatial resolution of ~13.5 km nadir horizontal resolution.

OMI is a nadir-viewing push broom ultraviolet-visible (UV-VIS) imaging spectrograph that measures backscattered radiances covering the 270–500 nm wavelength range (Levelt et al., 2006 a, b) and captures the absorption features of the
ozone Hartley and Huggins bands that are clearly present in the 270–310 nm (mainly for stratospheric ozone information) and 310–330 nm (mainly for tropospheric ozone information) spectral regions. The ground pixel size of OMI measurements at nadir position is about 13 km (along the ground track of spacecraft) × 24 km (across the track) when using the spectral radiances 310–330 nm. Since 2009, row anomaly and stray light issues have affected the quality of some OMI pixels (Huang et al., 2017; Schenkeveld et al., 2017; Levelt et al., 2018). Following 2009, for retrieval, the MUlti-SpEctra, MUlti-SpEcies,
MUlti-SEnsors (MUSES) algorithm uses the measured radiances from the OMI "healthy" off-nadir pixels and the corresponding collocated AIRS measurements.

The World Ozone and Ultraviolet radiation Data Centre (WOUDC, http://www.woudc.org) ozonesonde measurements provide in-situ data from the surface to the stratosphere (about 35 km) with vertical resolution of ~150 m and accuracy of 5% (Witte et al., 2017, 2018; WMO/GAW, 2017). These data fill a critical need for the validation of ozone profiles
measured by space-borne remote sensing instruments (Thompson et al., 2017). The ozonesonde sensor has a dilute solution of potassium iodide to produce a weak electrical current proportional to the ozone concentration of the sampled air (Komhyr et al., 1995). To examine the performances of remote sensing measurements, we applied the following coincidence criteria to determine sonde-AIRS+OMI and sonde-TES pairs: (1) mean cloud optical depth < 2.0, (2) cloud fraction within OMI field of view < 30%, (3) both satellite ground pixel-sonde distances < 300 km, (4) solar zenith angle < 80°,  and (5) daytime
measurements with a time difference < 4 hour. Using these criteria for the 2006 timeframe, we obtained 424 sonde-AIRS+OMI triads and 556 sonde-TES measurement pairs.

## 3 Retrieval algorithms and retrieval characteristics

The joint AIRS+OMI ozone profile is produced from the MUSES retrieval algorithm, crafted to accommodate multiple instruments including joint TES+OMI $O_3$ retrievals (Fu et al., 2013), joint CrIS+TROPOMI carbon monoxide (CO)
profiling (Fu et al., 2016), joint TES+Microwave Limb Sounder (MLS) CO retrievals (Luo et al., 2013), and AIRS $CH_4$, HDO, $H_2O$, and CO retrievals (Worden et al., 2018; Kulawik et al., 2018). These atmospheric composition products, with characteristics of vertical resolution and uncertainty similar to TES standard level 2 data, have the potential to extend the Aura TES atmospheric composition earth science data records (ESDRs), continuing the climate and air quality science enabled by TES measurements. The development of the MUSES algorithm leverages a suite of existing atmospheric composition retrieval
algorithms, especially forward radiative transfer models, including the Earth Limb and Nadir Operational Retrieval (ELANOR) of the TES operational algorithm (Worden et al., 2004; Clough et al., 2006; Kulawik et al., 2006a, b; Bowman et al., 2006; Eldering et al., 2008) for simulation of TIR radiances and Jacobians (Fu et al., 2013, 2016); the U.S. Smithsonian Astrophysical Observatory (SAO) OMI OZone PROFile (RROFOZ) algorithm (Liu et al., 2010 a, b) for simulation of UV radiances and




Jacobians of Hartley and Huggins bands (Fu et al., 2013; Worden et al., 2013); and the full physical OCO-2 algorithm (O'Dell et al., 2012, 2018; Connor et al., 2016; Crisp et al., 2012, 2017; Eldering et al., 2017) for simulation of short wavelength infrared radiances and Jacobians (Fu et al., 2016).

### 3.1 Joint AIRS+OMI ozone profile retrievals

5     The retrieval methodology is based on the optimal estimation (OE) method (Rodgers, 2000), which minimizes the differences between observed and measured radiances subject to a priori knowledge, i.e., mean and covariation of the atmospheric-cloud-surface state, to infer the "optimal" or maximum a posterior (MAP). Numerically, the MAP state vector $\hat{\mathbf{x}}$, which represents the concentration of atmospheric trace gases and ancillary parameters, is computed by minimizing the following cost function with respect to $\mathbf{x}$:

$$C(\mathbf{x}) = \|\mathbf{x} - \mathbf{x_a}\|^2_{\mathbf{S_a^{-1}}} + \|\mathbf{L_{obs}} - \mathbf{L_{sim}}\|^2_{\mathbf{S_\epsilon^{-1}}} . \qquad (1)$$

Equation (1) is a sum of quadratic functions representing a weighted Euclidean norm ($\|\mathbf{b}\|^2_a = \mathbf{b^T a b}$), with the first term accounting for the difference between the retrieval vector $\mathbf{x}$ and a priori state $\mathbf{x_a}$, inversely weighted by the a priori covariance matrix $\mathbf{S_a}$, and with the second term representing the difference between the observed $\mathbf{L_{obs}}$ and simulated $\mathbf{L_{sim}}$ radiance spectra inversely weighted by the measurement error covariance matrix $\mathbf{S_\epsilon}$.

15    Under the assumption that measurement error between AIRS and OMI is uncorrelated, Eq. (1) can be written as

$$C(\mathbf{x}) = \|\mathbf{x} - \mathbf{x_a}\|^2_{\mathbf{S_a^{-1}}} + \underbrace{\left\|\mathbf{L_{obs\_AIRS}} - \mathbf{L_{sim\_AIRS}}\right\|^2_{\mathbf{S_{\epsilon\_AIRS}^{-1}}}}_{\text{AIRS}} + \underbrace{\left\|\mathbf{L_{obs\_OMI}} - \mathbf{L_{sim\_OMI}}\right\|^2_{\mathbf{S_{\epsilon\_OMI}^{-1}}}}_{\text{OMI}} . \qquad (2)$$

The joint retrieval algorithm iteratively updates the state vector based upon a trust-region Levenberg–Marquardt (LM) optimization algorithm (Moré, 1977, Bowman et al., 2006) to minimize the cost function in Eq. (2):

$$\mathbf{x_{i+1}} = \mathbf{x_i} + \left[ \gamma_i \mathbf{W^T W} + \mathbf{S_a^{-1}} + \underbrace{\mathbf{K_{AIRS}^T S_{\epsilon\_AIRS}^{-1} K_{AIRS}}}_{\text{AIRS}} + \underbrace{\mathbf{K_{OMI}^T S_{\epsilon\_OMI}^{-1} K_{OMI}}}_{\text{OMI}} \right]^{-1} \times \left[ \mathbf{S_a^{-1}}(\mathbf{x_a} - \mathbf{x_i}) + \underbrace{\mathbf{K_{AIRS}^T S_{\epsilon\_AIRS}^{-1} \Delta L_{AIRS}}}_{\text{AIRS}} + \right.$$

$$\left. \underbrace{\mathbf{K_{OMI}^T S_{\epsilon\_OMI}^{-1} \Delta L_{OMI}}}_{\text{OMI}} \right], \qquad (3)$$

Where, the parameter $\gamma_i$ is called the LM parameter, $\mathbf{W}$ is a nonzero scaling matrix, $\mathbf{K_{instrument}}$ is the Jacobian matrix representing instrument sensitivity of spectral radiances to the atmospheric state, and $\Delta\mathbf{L}$ is the difference between observed and simulated spectral radiances.

        To simulate TIR spectral radiances $\mathbf{L}$ and Jacobians $\mathbf{K}$ in TIR and UV spectral regions (Table 1), the joint AIRS+OMI
25    retrieval adopts the forward models of the joint TES+OMI retrievals (Fu et al., 2013) with necessary revisions to incorporate the AIRS specifications (spectral range, signal-to-noise ratios (SNR), viewing geometry, and spectral response function) (Pagano et al., 2003; Strow et al., 2003).



**Table 1: Spectral regions used in ozone retrievals.**

| Case Selection[a] | Spectral Data | Frequency Start (cm$^{-1}$) | Frequency End (cm$^{-1}$) | Resolving Power | Atmospheric Species |
|---|---|---|---|---|---|
| AIRS+OMI, AIRS | | 985.10 | 1,031.24 | | $H_2O$, $O_3$, $CO_2$ |
| AIRS+OMI, AIRS | | 1,042.76 | 1,048.58 | | $H_2O$, $O_3$, $CO_2$ |
| AIRS+OMI, AIRS | | 1,068.98 | 1,071.38 | | $H_2O$, $O_3$, $CO_2$ |
| AIRS+OMI, AIRS | | 1,108.88 | 1,112.06 | | $H_2O$, $O_3$, $CO_2$ |
| AIRS+OMI, AIRS | AIRS level 1B Version 5 data[b] | 1,224.10 | 1,227.88 | 1,200 | $H_2O$, HDO, $O_3$, $CO_2$, $CH_4$, $N_2O$ |
| AIRS+OMI, AIRS | | 1,259.38 | 1,261.42 | | $H_2O$, HDO, $O_3$, $CO_2$, $CH_4$, $N_2O$ |
| AIRS+OMI, AIRS | | 1,265.92 | 1,267.06 | | $H_2O$, HDO, $O_3$, $CO_2$, $CH_4$, $N_2O$ |
| AIRS+OMI, AIRS | | 1,269.46 | 1,270.54 | | $H_2O$, HDO, $O_3$, $CO_2$, $CH_4$, $N_2O$ |
| AIRS+OMI, AIRS | | 1,311.70 | 1,315.36 | | $H_2O$, HDO, $O_3$, $CO_2$, $CH_4$, $N_2O$ |
| AIRS+OMI, AIRS | | 1,315.72 | 1,317.82 | | $H_2O$, HDO, $O_3$, $CO_2$, $CH_4$, $N_2O$ |
| AIRS+OMI, OMI | OMI level 1B Version 3 data[c] | 270.00 | 310.00 | 460 | $O_3$ |
| AIRS+OMI, OMI | | 310.00 | 330.00 | 800 | $O_3$ |

[a] The parameters are included in the retrievals for different cases (AIRS only, OMI only, and joint AIRS+OMI).
[b] AIRS single footprint infrared geolocated and calibrated radiance data (Aumann et al., 2003) are used directly rather than level 2 cloud-cleared spectra, which are calculated using nine adjacent AIRS infrared footprints. Using single-footprint spectra improves the performance of horizontal resolution of the AIRS retrieval from ~45 to ~13.5 km at nadir, leading to improved representation of horizontal details (Irion et al., 2018).
[c] Retrievals normalized radiances (i.e., $I_{Earthshine}$/ $I_{solar\_iiradiance}$) were used in the retrievals. OMI level 1B global geolocated earthshine radiance ($I_{Earthshine}$) and solar irradiances ($I_{solar\_iiradiance}$) (Dobber et al., 2006a, b; Van den Oord et al., 2006).

The joint AIRS+OMI retrievals start with the list of the fitting parameters, a priori values, and a priori variance shown in Table 2. In addition to the initial guess for the trace gas concentration ($O_3$, $H_2O$, and $CO_2$), the initial guess for auxiliary parameters used in the simulation of AIRS radiances (including temperature profile, surface temperature and emissivity, cloud extinction and cloud top pressure) are also retrieved from AIRS radiances in order to take into account their spectral signatures in the $O_3$ spectral regions. The joint AIRS+OMI algorithm incorporated a suite of treatments in order to optimize the spatial resolution, retrieval stability, data throughput, and consistency to TES data products (version 6): (1) single-footprint AIRS level 1B radiances in the retrievals (Irion et al., 2018), which leads to a footprint nine times smaller in area than the AIRS version 6 operational algorithm (Susskind et al., 2003 and 2014); (2) global infrared land surface emissivity database from the University of Wisconsin-Madison (UOW-M) (Seemann et al., 2007), which improves clear land throughput by 4.5%; (3) an initial guess refinement step of cloud fraction prior to the step of joint AIRS+OMI ozone retrievals to reduce the impacts of cloud interference on trace gas retrievals; (4) a priori constraint vector and matrix identical to the TES version 6 operational algorithm to obtain uncertainty estimates consistent with TES data products; (5) an updated a priori and initial guess information of atmospheric temperature profiles taken from the Goddard Earth Observing System Model, Version 5 (GEOS-5) (Rienecker et al., 2008) for AIRS TIR temperature profile retrievals; (6) updated a priori ozone estimated from the Model for OZone and Related chemical Tracers (MOZART)-4 (Emmons et al., 2010); and (7) HIgh-resolution TRANsmission (HITRAN) 2012 (Rothman et al., 2013) spectroscopic parameters and a priori information of water vapor, the primary interfering species in TIR ozone measurements jointly retrieved with ozone.



**Table 2: List of parameters in state vector.**

| Case Selection[a] | Fitting Parameters | Number of Parameters | A Priori | A Priori Uncertainty |
|---|---|---|---|---|
| AIRS+OMI, AIRS, OMI | $O_3$ at each pressure level | 25 | MOZART-4[b] | MOZART-3 ~10–40% |
| AIRS+OMI, AIRS | $H_2O$ at each pressure level | 16 | GEOS-5[c] | NCEP[d] ~30% |
| AIRS+OMI, AIRS | Surface temperature | 1 | GEOS-5 | 0.5 K |
| AIRS+OMI, AIRS | Surface emissivity[e] | 23 | UOW-M[f] | ~0.006 |
| AIRS+OMI, AIRS | Cloud extinction[g] | 11 | Initial BT difference | 300% |
| AIRS+OMI, AIRS | Cloud top pressure[g] | 1 | 500 mbar | 100% |
| AIRS+OMI, OMI | UV1 surface albedo | 1 | OMI climatology[h] | 0.05 |
| AIRS+OMI, OMI | UV2 surface albedo (zero order)[i] | 2 | OMI climatology | 0.05 |
| AIRS+OMI, OMI | UV2 surface albedo (first order)[i] | | 0 | 0.01 |
| AIRS+OMI, OMI | UV1, UV2 ring scaling factors | 2 | 1.9 | 1.0 |
| AIRS+OMI, OMI | UV1, UV2 radiance/irradiance wavelength shifts | 2 | 0 | 0.02 nm |
| AIRS+OMI, OMI | UV1, UV2 radiance/$O_3$ cross-section wavelength shifts | 2 | 0 | 0.02 nm |
| AIRS+OMI, OMI | Cloud fraction[j] | 1 | Derived from 347 nm | 0.05 |

[a] The parameters are included in the retrievals for different cases (AIRS only, OMI only, and joint AIRS+OMI).
[b] Model for OZone and Related chemical Tracers (MOZART)-4 (Emmons et al., 2010)
[c] Goddard Earth Observing System, version 5 (GEOS-5) (Rienecker et al., 2008)
[d] National Center for Environmental Prediction (NCEP) reanalysis (Kalnay et al., 1996)
[e] Retrievals over land; spectral surface emissivity is factored in.
[f] Global infrared land surface emissivity database at University of Wisconsin-Madison (UOW-M) (Seemann et al., 2007).
[g] For cloud treatment in TIR spectral region, we adopt the approach used in the TES level-2 full-physics retrieval algorithm (Kulawik et al., 2006b; Eldering et al., 2008). Gaussian parameters represent the total optical depth, peak altitude, and profile width.
[h] The surface reflectance climatology was constructed using 3 year of OMI measurements obtained between 2004 and 2007 (Kleipool et al., 2008).
[i] The surface is assumed to be Lambertian with a variable slope in wavelength to the albedo, such that the albedo can vary linearly across the spectral band.
[j] For cloud treatment in UV spectral region, we adopt the approach used in the TES+OMI retrieval algorithm (Fu et al., 2013) by adding in an initial guess refinement step for retrieving the cloud fraction prior to joint AIRS+OMI ozone retrievals.

## 3.2 Retrieval characteristics of TES, AIRS, OMI, and joint AIRS/OMI

For moderately non-linear problems, the estimated state can be written as the linear expression (Worden et al., 2007a):

$$\hat{\mathbf{x}} = \mathbf{x}_a + \mathbf{A}[\mathbf{x}_{true} - \mathbf{x}_a] + \mathbf{G}\varepsilon + \delta_{cs}, \tag{4}$$

where $\mathbf{x}_a$ is the a priori constraint vector, $\mathbf{A}$ is the averaging kernel matrix whose rows represent the sensitivity of the retrieval to the true state, $\mathbf{x}_{true}$ is the true state vector, $\varepsilon$ is the spectral noise, and $\mathbf{G}$ is the gain matrix, which can be written as $\mathbf{G} = (\mathbf{K}^T\mathbf{S}_\epsilon^{-1}\mathbf{K} + \mathbf{S}_a^{-1})^{-1}\mathbf{K}^T\mathbf{S}_\epsilon^{-1}$. The "cross-state" error, $\delta_{cs}$, is incurred from retrieving multiple parameters (e.g., water vapor, surface temperature, cloud extinction and cloud top pressure in TIR, cloud fraction in UV, surface albedo, and wavelength shifting parameters).

The use of OE in the MUSES algorithm also provides the averaging kernel and uncertainty matrices for each sounding needed for trend analysis, climate model evaluation, and data assimilation. Based on the optimal estimation theory, the averaging kernel matrix ($\mathbf{A}$) and total error covariance matrix ($\mathbf{S}$) can be calculated as follows:



$$A = GK, \tag{5}$$

$$S = \underbrace{(I - A)S_a(I - A^T)}_{\text{Smoothing Error}} + \underbrace{GS_\varepsilon G^T}_{\substack{\text{Satellite Instrument} \\ \text{Measurement} \\ \text{Error}}} + \underbrace{A_{cs}S_{cs}A_{cs}^T}_{\substack{\text{Cross} \\ \text{State} \\ \text{Error}}} , \tag{6}$$

$$\underbrace{\phantom{GS_\varepsilon G^T + A_{cs}S_{cs}A_{cs}^T}}_{\text{Satellite Instrument Observation Error}}$$

where $\mathbf{I}$ is the identity matrix; $\mathbf{S}_a$ is the a priori covariance matrix of the full retrieved state containing both atmospheric and auxiliary parameters; $\mathbf{S}_\varepsilon$ is the measurement noise covariance of both TIR and UV radiances.

The trace of the averaging kernel matrix ($\mathbf{A}$) gives the number of independent pieces of information in the vertical profile, or, the degrees of freedom for signal (DOFS) (Rodgers, 2000). A larger DOFS value indicates a better vertical sensitivity. Figure 1 shows sample averaging kernel matrices for TES, AIRS, OMI, and joint AIRS+OMI transect observations over the western United States on August 23, 2006. The joint AIRS+OMI and TES retrievals show similar capability for resolving the lower/upper troposphere (tropospheric DOFS: 1.64 for TES; 1.55 for joint AIRS+OMI). Both AIRS and OMI

tropospheric DOFS are ~1 – capable of estimating the tropospheric columns but lacking vertical sensitivity in the troposphere.

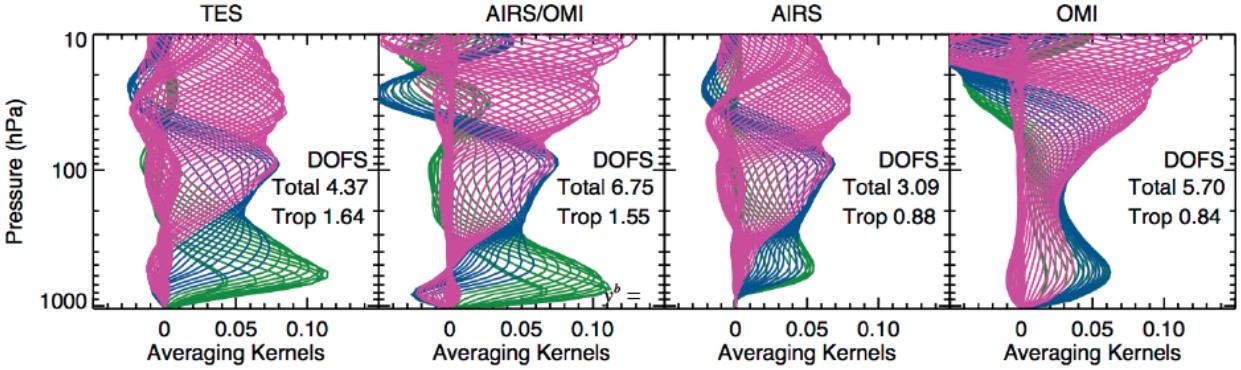

**Figure 1: Averaging kernels of collocated measurements of TES (version 6), joint AIRS+OMI, AIRS alone, and OMI alone over California, USA on August 23, 2006. The green, blue, and magenta curves in four panels indicate the averaging kernels in the pressure range of surface-400 hPa, 400 to 100 hPa, and above 100 hPa, accordingly.**

## 15    4 Validation of joint AIRS+OMI data

An initial comparison between TES, AIRS, OMI, and AIRS+OMI is shown by a transect from ~6°N to 55°N taken on August 23, 2006 (Fig. 2A) and processed through the MUSES algorithm. The tropospheric ozone concentration profiles of joint AIRS+OMI retrievals show better agreement with TES data (Fig. 2G, green curve; mean differences < 2% from surface to 400 hPa, and < 5% from 400 hPa to 100 hPa), than the retrievals for both AIRS and OMI alone (Fig. 2G, blue curve for AIRS, purple

curve for OMI). The joint retrievals improve the agreement due to the increased vertical sensitivity in comparison to each instrument alone as the multispectral retrievals have the advantage of obtaining the information from multiple physical regimes, including the vertical distribution via thermal emissions, pressure-temperature dependent spectral line broadenings and absorption cross sections, and wavelength-altitude dependent atmospheric scattering events via UV radiances.



Further evaluation of the joint AIRS+OMI O₃ retrievals are shown in two modes: Global Survey (GS) and REgional mapping (RE). The GS mode provides profile data with a temporal/spatial sampling similar to TES GS, while RE mode processes all available AIRS+OMI measurements over a region, specifically in this case we have considered the Korean peninsula during the 2016 KORUS-AQ campaign (Miyazaki et al., 2018). The global joint AIRS+OMI retrievals have been compared to the well-validated TES data (Sec. 4.1) and high accuracy in-situ global ozonesonde measurements (Sec. 4.2) to quantify the performance of this multispectral tropospheric ozone profile data product. These comparisons were made using measurements in 2006, when neither the TES instrument degradation nor OMI row anomaly played a role.

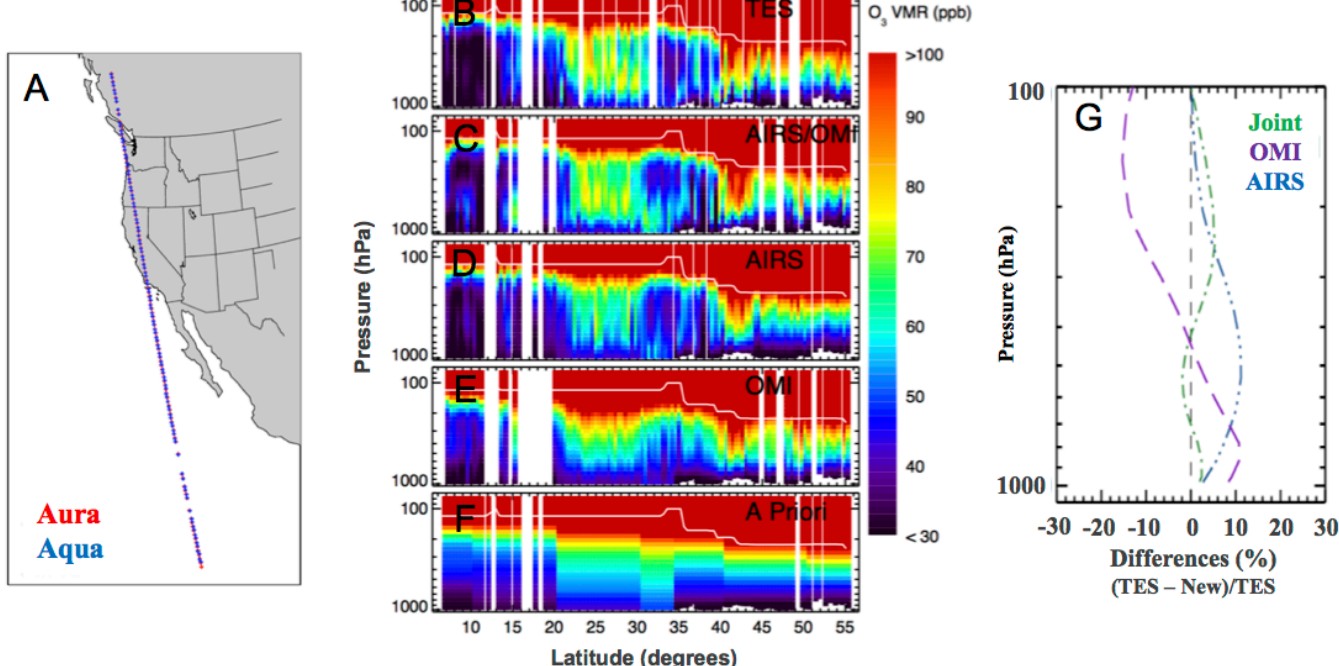

**Figure 2: Collocated ozone (O₃) measurements from A-Train nadir viewing spectrometers over the western United States on August 23, 2006. (A) geolocation of 110 TES-AIRS-OMI triads (spatial-temporal differences ~8 km; ~16 minutes); (B) vertical profile of TES O₃ volume mixing ratio (VMR) data (version 6) with a unit of parts per billion (ppb); (C) joint AIRS+OMI retrievals; (D) AIRS alone; (E) OMI alone; (F) a priori used in retrievals; (G) averaged percentage differences of retrieved O₃ profiles in comparison to TES O₃ data (version 6): TES vs. joint AIRS+OMI (green dash-dot), TES vs. AIRS alone (blue), and TES vs. OMI alone (purple dash). The white curves in the panels of B–F indicate the tropopause pressure taken from the Goddard Earth Observing System Model, Version 5.**

## 4.1 Comparison to the TES data

Both TES and joint AIRS+OMI 2006 ozone profile data were screened prior to the comparison using (1) the ''species retrieval quality'' – a master quality flag available in the level 2 product files; (2) the retrieved cloud effective TIR optical depth (removed when OD > 2.0); (3) cloud fraction within field of view (excluded when effective cloud fraction in OMI > 30%); (4) solar zenith angle (SZA; excluded when SZA > 80°). We excluded profiles with thick clouds in the field of view because these obscure the infrared emission from the lower troposphere, which greatly reduces the satellite sensitivity of both IR and UV radiances; only daytime scenes are included since OMI measurements depend on the sunlight.





Joint AIRS+OMI global tropospheric O₃ retrievals (Figs. 3A1–3) show good agreement with TES data, as shown in
Figs. 3B1–3. Both datasets are significantly different from the a priori and capture the synoptic ozone patterns such as the
midlatitude Atlantic and the biomass burning events (e.g., Southern Africa). Results for the remaining months of 2006 are
available in supplement Figures S1–S11. The correlation coefficients of joint AIRS+OMI and TES version 6 data (Table 3)

are greater than 0.71 and up to 0.92 for all months across the troposphere where the mean and root mean square (RMS) of the
differences of two data sets (Table 3) are well within the estimated total uncertainty.

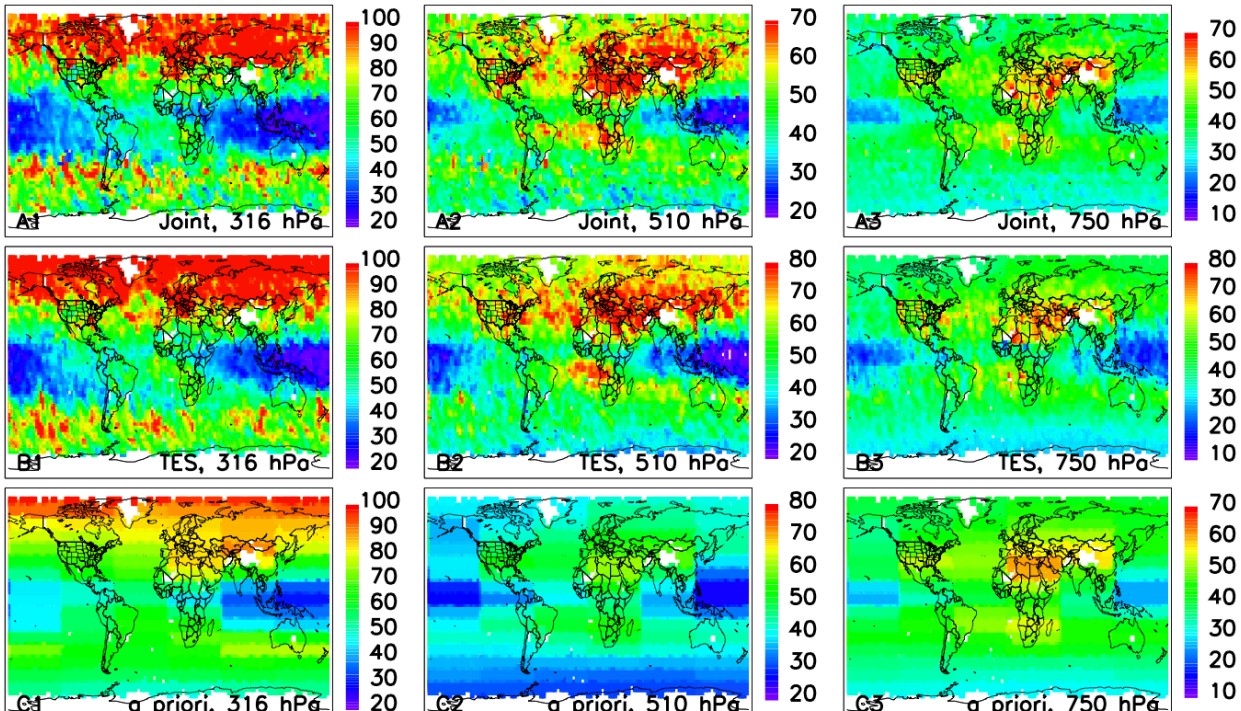

**Figure 3: Global maps of monthly averaged ozone (O₃) volume mixing ratio (VMR) with a unit of ppb. The A-Train measurements
in August 2006 were used in creating these global maps. Comparison of Joint AIRS+OMI (top row, A), TES (middle row, B), and a**
**priori (bottom row C) ozone VMR for the pressure level of 316, 510 and 750 hPa (columns left, middle, right), respectively. All data
have been gridded to 2.5° × 2.5° cells. Results for the remaining months of 2006 are available in supplement Figures S1–S11.**

The characteristics of the joint AIRS+OMI retrievals, in terms of vertical sensitivity and estimated uncertainty
characteristics, are similar to those of TES data. The DOFS of global tropospheric ozone retrievals show distributions similar
to TES data (Figs. 4 panels A2 and B2, supplement Figures S12–S22 for the remaining months of 2006). Also, the estimated

observation and total uncertainties of joint AIRS+OMI retrievals (black curves of Fig. 5) show peaks and widths equivalent
to that of TES data products (green curves of Fig. 5) across troposphere over the globe (supplement Figures S23–S33 for the
remaining months of 2006): i.e., the peak of the estimated observation errors, which are the sum of second and third terms in
Eq, (6), reside in the range of 6–8% (or ~3 ppb) for the joint AIRS+OMI retrievals – equivalent to the observation uncertainty
of 5–7% (or ~2–3 ppb) from TES data across the troposphere. Finally, the joint AIRS+OMI retrievals have total uncertainties

equivalent to TES data (within 3% agreement) over the globe.



**Table 3: Comparisons between joint AIRS/OMI and TES gridded (2.5° × 2.5°) global survey measurements of ozone concentration at three pressure levels (316 hPa, 510 hPa, and 750 hPa) for year 2006.**

| 316 hPa | | Jan | Feb | Mar | Apr | May | Jun | Jul | Aug | Sep | Oct | Nov | Dec |
|---|---|---|---|---|---|---|---|---|---|---|---|---|---|
| Pearson Correlation Coefficient | | 0.83 | 0.84 | 0.85 | 0.84 | 0.84 | 0.84 | 0.84 | 0.82 | 0.74 | 0.74 | 0.71 | 0.78 |
| Differences (TES-AIRS+OMI) | Mean (ppb) | 8.3 | 8.5 | 7.3 | 6.9 | 8.1 | 6.0 | 4.8 | 2.8 | 1.4 | 1.9 | 3.4 | 5.4 |
| | RMS (ppb) | 16.3 | 20.5 | 21.5 | 21.6 | 22.6 | 19.8 | 17.8 | 15.6 | 16.2 | 14.9 | 13.4 | 13.2 |
| | Mean (%) | 12.9 | 11.6 | 9.8 | 7.3 | 7.3 | 5.0 | 3.6 | 2.3 | 0.1 | 1.9 | 4.3 | 7.7 |
| | RMS (%) | 24.1 | 26.5 | 24.2 | 25.7 | 24.7 | 23.8 | 22.1 | 22.2 | 23.2 | 23.7 | 21.2 | 20.8 |
| Total Uncertainty | AIRS+OMI O$_3$ (%) | 28.8 | 28.8 | 28.6 | 28.9 | 28.5 | 28.0 | 27.9 | 27.5 | 27.8 | 28.2 | 28.8 | 28.9 |
| | TES V6 O$_3$ (%) | 22.7 | 22.6 | 22.5 | 23.0 | 22.9 | 22.1 | 22.1 | 22.2 | 22.5 | 22.3 | 22.8 | 22.9 |
| 510 hPa | | Jan | Feb | Mar | Apr | May | Jun | Jul | Aug | Sep | Oct | Nov | Dec |
| Pearson Correlation Coefficient | | 0.81 | 0.84 | 0.87 | 0.88 | 0.89 | 0.86 | 0.86 | 0.82 | 0.74 | 0.74 | 0.71 | 0.79 |
| Differences (TES-AIRS+OMI) | Mean (ppb) | 3.3 | 2.6 | 2.9 | 3.3 | 3.6 | 4.1 | 4.1 | 4.2 | 3.9 | 3.2 | 3.1 | 3.5 |
| | RMS (ppb) | 7.7 | 8.3 | 8.6 | 8.9 | 9.2 | 9.5 | 8.7 | 8.2 | 8.7 | 8.2 | 7.4 | 7.1 |
| | Mean (%) | 6.5 | 3.8 | 4.9 | 4.2 | 4.2 | 4.5 | 4.6 | 5.6 | 4.7 | 4.2 | 5.2 | 6.6 |
| | RMS (%) | 16.3 | 18.2 | 17.3 | 18.2 | 16.4 | 17.0 | 16.5 | 15.0 | 16.5 | 16.6 | 15.4 | 15.4 |
| Total Uncertainty | AIRS+OMI O$_3$ (%) | 22.5 | 22.4 | 22.5 | 22.8 | 23.0 | 22.8 | 22.7 | 22.6 | 22.4 | 22.3 | 22.4 | 22.5 |
| | TES V6 O$_3$ (%) | 20.4 | 20.4 | 20.1 | 20.1 | 20.1 | 19.5 | 19.6 | 19.7 | 19.6 | 19.4 | 19.9 | 20.3 |
| 750 hPa | | Jan | Feb | Mar | Apr | May | Jun | Jul | Aug | Sep | Oct | Nov | Dec |
| Pearson Correlation Coefficient | | 0.92 | 0.89 | 0.90 | 0.90 | 0.90 | 0.83 | 0.82 | 0.80 | 0.74 | 0.76 | 0.87 | 0.92 |
| Differences (TES-AIRS+OMI) | Mean (ppb) | 0.4 | -0.8 | 0.4 | 1.2 | 1.6 | 2.2 | 2.3 | 3.2 | 3.3 | 2.0 | 1.5 | 1.4 |
| | RMS (ppb) | 5.2 | 6.4 | 6.7 | 7.0 | 6.4 | 7.1 | 6.4 | 6.3 | 6.9 | 6.1 | 5.2 | 4.8 |
| | Mean (%) | 0.2 | -4.1 | 0.6 | 0.3 | 1.3 | 2.3 | 2.9 | 5.4 | 5.4 | 2.9 | 2.9 | 3.0 |
| | RMS (%) | 14.7 | 19.3 | 19.3 | 19.8 | 15.9 | 17.4 | 16.0 | 14.6 | 16.8 | 16.1 | 14.3 | 13.4 |
| Total Uncertainty | AIRS+OMI O$_3$ (%) | 23.9 | 22.9 | 22.4 | 22.9 | 24.1 | 24.7 | 24.6 | 24.2 | 23.9 | 24.2 | 23.9 | 24.1 |
| | TES V6 (%) | 24.2 | 23.8 | 23.1 | 23.3 | 24.0 | 24.0 | 24.0 | 23.6 | 23.4 | 23.4 | 23.5 | 23.8 |
| Number of Global Survey | AIRS+OMI | 16 | 13 | 14 | 15 | 16 | 15 | 15 | 16 | 16 | 15 | 15 | 16 |
| | TES | 15 | 13 | 16 | 14 | 15 | 15 | 16 | 14 | 9 | 16 | 15 | 15 |

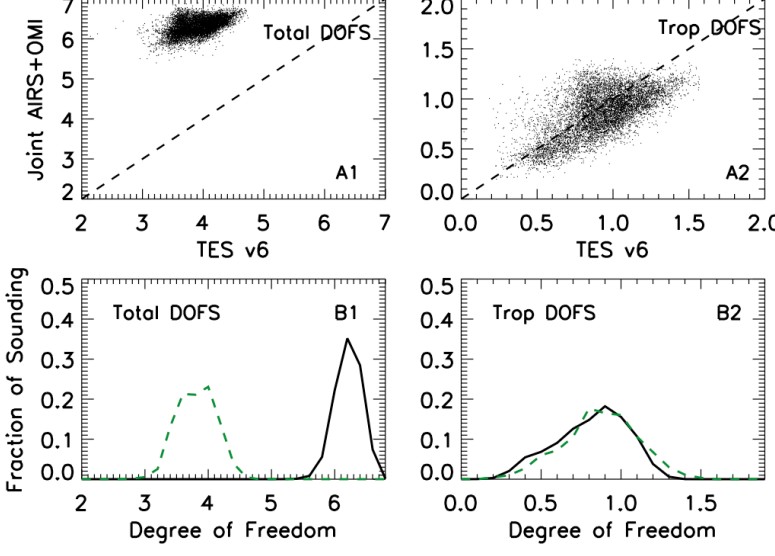





**Figure 4: DOFS for O$_3$ over globe shown in Fig. 3. Here, we used the A-Train measurements from August 2006. Results for the remaining months of 2006 are available in supplement Figures S12–S22. (A1) total DOFS; (A2) tropospheric DOFS; (B1) histogram of total DOFS: joint AIRS+OMI (black line) and TES version 6 (green dash); and (B2) histogram of tropospheric DOFS joint AIRS+OMI (black line) and TES version 6 (green dash).**

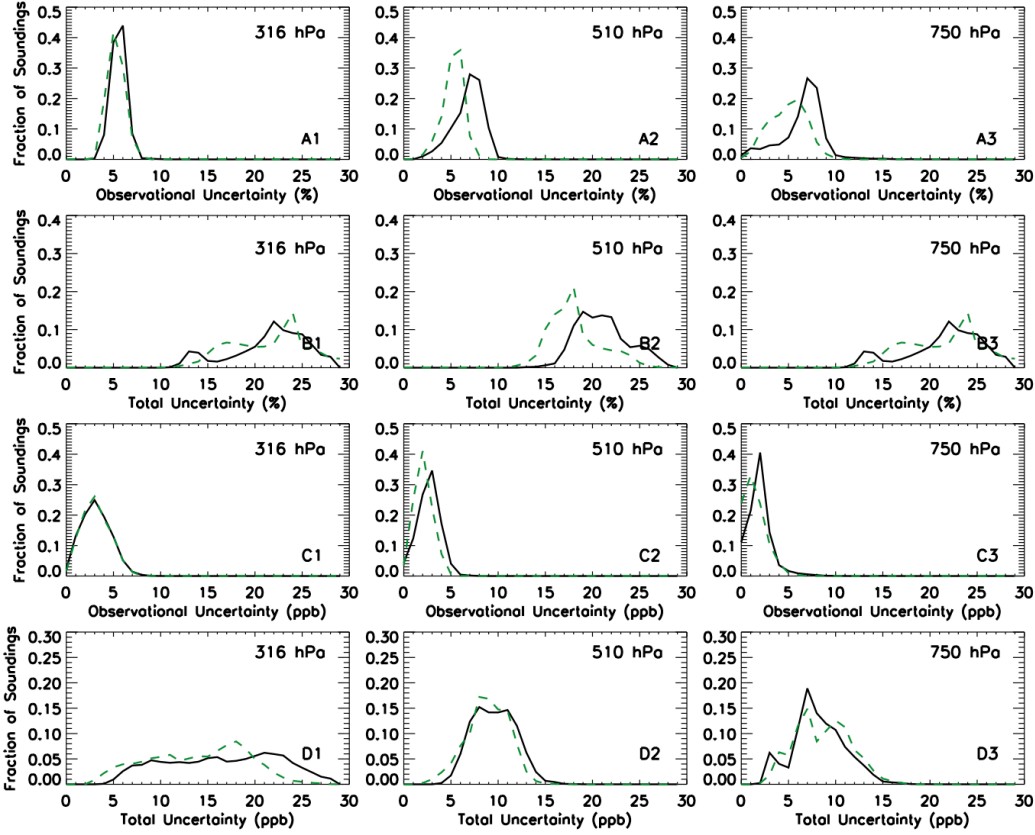

**Figure 5: Estimated (predicted) uncertainty of retrieved global O$_3$ concentration shown in Fig. 3. Here, we used the A-Train measurements from August 2006. Results for the remaining months of 2006 are available in supplement Figures S23–S33. (A1–A3) observational uncertainty; (B1–B3) total uncertainty; (C1–C3) observational uncertainty in ppb; and (D1–D3) total uncertainty in ppb. Joint AIRS+OMI data are shown in black line, and TES version 6 data are shown in green dash.**

10 ## 4.2 Comparison to ozonesonde measurements

We identified 424 sonde-AIRS+OMI triads and 556 sonde-TES pairs following the coincidence criteria in Sec. 2. Following Worden et al. (2007a), satellite observation operators $\mathbf{H}(\mathbf{x}_a, \mathbf{A})$ defined in the equation for joint AIRS+OMI and TES were applied to the in-situ ozonesonde profiles accounting for known bias and precision. As a result, the expected covariance matrix of the differences between the satellite retrievals and ozonesonde measurements smoothed by instrument

15 averaging kernels can be written as similarly to Eq. (6) (Worden et al., 2007a; Fu et al., 2013):

$$E[(\hat{\mathbf{x}} - \hat{\mathbf{x}}_{sonde})(\hat{\mathbf{x}} - \hat{\mathbf{x}}_{sonde})^T] = \underbrace{\mathbf{A S}_{sonde}\mathbf{A}^T}_{\substack{\text{Ozonesonde}\\\text{Measurement}\\\text{Error}}} + \underbrace{\mathbf{G S}_\varepsilon\mathbf{G}^T}_{\substack{\text{Satellite Instrument}\\\text{Measurement}\\\text{Error}}} + \underbrace{\mathbf{A}_{cs}\mathbf{S}_{cs}\mathbf{A}_{cs}^T}_{\substack{\text{Cross}\\\text{State}\\\text{Error}}} + \underbrace{\mathbf{G S}_{\varepsilon r}\mathbf{G}^T}_{\substack{\text{Remaining}\\\text{Radiance}\\\text{Calibration}\\\text{Error}}} + \underbrace{\mathbf{S}_{SS}}_{\substack{\text{Sonde}-\text{Satellite}\\\text{Temporal Spatial}\\\text{Sampling}}} \quad (7)$$

$$\underbrace{\phantom{\mathbf{G S}_\varepsilon\mathbf{G}^T + \mathbf{A}_{cs}\mathbf{S}_{cs}\mathbf{A}_{cs}^T}}_{\text{Satellite Instrument Observation Error}}$$



Eq. (7) indicates that the error covariance matrix is not biased by the a priori $\mathbf{x}_a$, and the biases of $O_3$ retrievals relative to ozonesondes are due to the errors of the sonde measurements $\mathbf{S}_s$, the random spectral noise $\mathbf{S}_\varepsilon$, the interfereing parameters in retrieval state vector $\mathbf{S}_{cs}$, the remaining radiometric calibration errors $\mathbf{S}_{\varepsilon r}$, or sonde-satellite temporal/spatial samplings $\mathbf{S}_{SS}$.

Figure 6 and Table 4 illustrate that both joint AIRS+OMI and TES data are in good agreement with ozonesonde measurements across seasonal variations in the troposphere. Here, the biases of ozone from remote-sensing measurements are within 3 ppb, 2 ppb, and 5 ppb for joint AIRS+OMI at three pressure levels (316, 510, and 750 hPa, respectively), while within 6 ppb, 4 ppb, and 3 ppb, respectively, for TES version 6 data. The biases of these satellite data show an improvement for all seasons when compared to a high bias of 3 to 10 ppb estimated for the TES tropospheric ozone data prior to version 6 via validation using ozonesonde measurements (Nassar et al., 2008; Boxe et al., 2010). Additionally, the RMS of the differences

are 10–17 ppb, 8–11 ppb, 7–9 ppb for the tropospheric ozone of joint AIRS+OMI retrievals, while 12–22 ppb, 8–15 ppb, 7–13 ppb for TES version 6 data, consistent with those reported by the existing TES validations. Overall comparisons of AIRS+OMI to ozonesondes yield similar biases and errors to matching comparisons between TES and sondes.

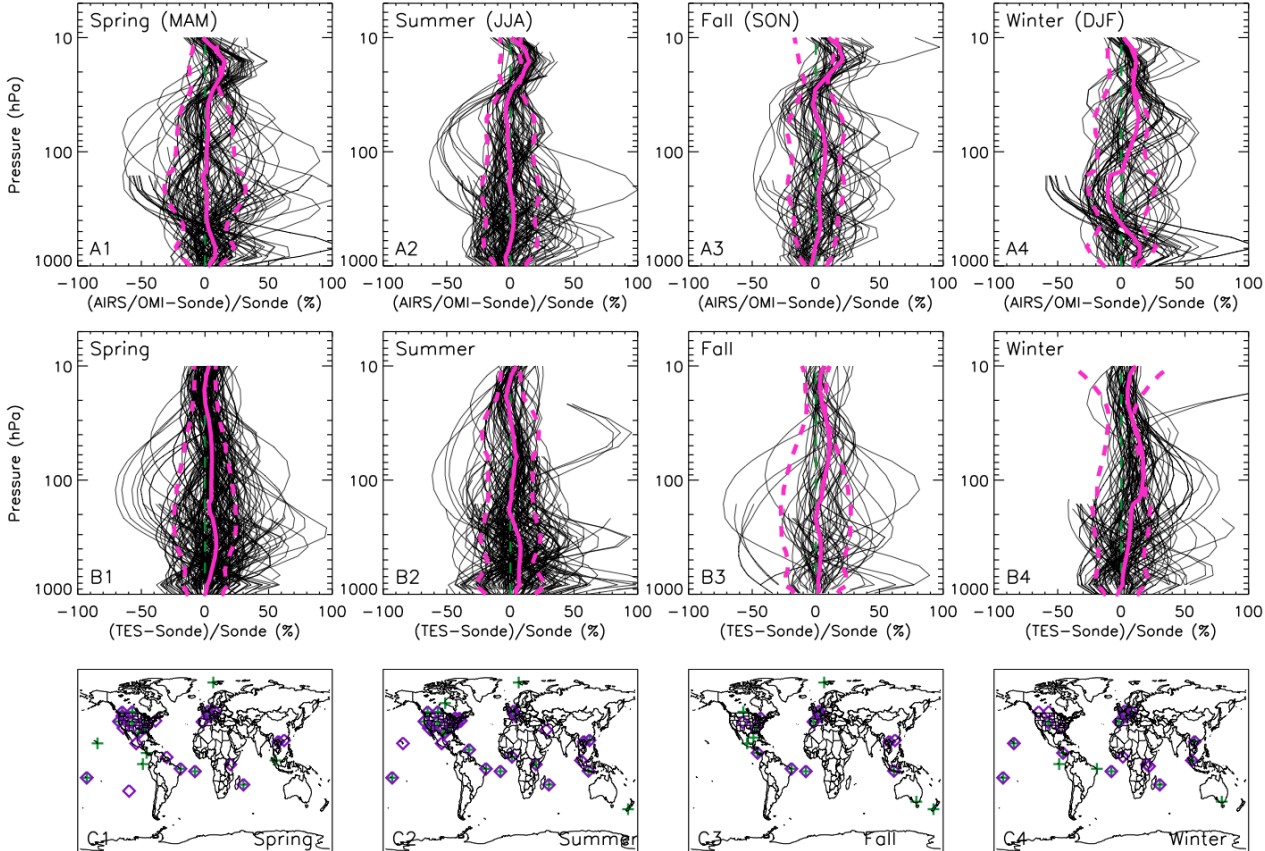

**Figure 6: Joint AIRS+OMI-sonde (A1–A4) and TES-sonde (B1–B4) percentage differences of measured ozone concentration for the**
15 **four seasons (months abbreviated in parentheses) over global. Individual profiles are shown in black, and the mean and 1 sigma standard deviation range are overlaid in solid magenta (mean) and dashed magenta lines. The profiles were plotted after removing cloudy scenes and flagged satellite (joint AIRS+OMI and TES) data. (A1–A4) joint AIRS+OMI vs. ozonesonde; (B1–B4) TES data**



**(version 6) vs. ozonesonde; (C1–C4) WOUDC sonde location that have coincident measurements with joint AIRS+OMI (green plus signs) and TES (purple diamonds).**

**Table 4: Comparisons between satellite remote sensing and ozone sonde in-situ measurements for 2006 at three pressure levels (316 hPa, 510 hPa, and 750 hPa).**

| | 316 hPa | Spring | | Summer | | Fall | | Winter | |
|---|---|---|---|---|---|---|---|---|---|
| | | AIRS+OMI | TES | AIRS+OMI | TES | AIRS+OMI | TES | AIRS+OMI | TES |
| | Mean (ppb) | 2.8 | 6.1 | 0.7 | 4.2 | 1.1 | -1.6 | -2.5 | 2.9 |
| | Mean (%) | 1.3 | 8.6 | 2.2 | 6.6 | 2.9 | 3.3 | -7.7 | 6.5 |
| | RMS (ppb) | 17.1 | 19.2 | 13.4 | 17.0 | 12.6 | 21.7 | 10.0 | 12.4 |
| | RMS (%) | 25.6 | 23.7 | 20.4 | 23.8 | 19.0 | 26.9 | 20.8 | 20.5 |
| | 510 hPa | Spring | | Summer | | Fall | | Winter | |
| | | AIRS+OMI | TES | AIRS+OMI | TES | AIRS+OMI | TES | AIRS+OMI | TES |
| Differences (Satellite – WOUC Sonde with Satellite Observation Operator Applied) | Mean (ppb) | 1.3 | 3.6 | -0.8 | 3.5 | 0.4 | 0.2 | 1.8 | 1.4 |
| | Mean (%) | 3.8 | 7.0 | 1.6 | 7.3 | 2.5 | 3.5 | 5.9 | 3.2 |
| | RMS (ppb) | 7.6 | 9.2 | 10.9 | 10.6 | 8.6 | 14.5 | 7.5 | 8.0 |
| | RMS (%) | 17.2 | 17.4 | 20.4 | 17.9 | 16.7 | 21.8 | 19.1 | 17.7 |
| | 750 hPa | Spring | | Summer | | Fall | | Winter | |
| | | AIRS+OMI | TES | AIRS+OMI | TES | AIRS+OMI | TES | AIRS+OMI | TES |
| | Mean (ppb) | 2.4 | 1.7 | -2.2 | 2.6 | -1.2 | 0.3 | 4.6 | 0.3 |
| | Mean (%) | 8.0 | 3.4 | -2.0 | 6.6 | -1.3 | 1.9 | 14.4 | 0.9 |
| | RMS (ppb) | 7.6 | 6.9 | 8.6 | 12.5 | 6.3 | 11.2 | 8.5 | 7.8 |
| | RMS (%) | 21.1 | 16.2 | 18.8 | 25.3 | 13.2 | 23.9 | 24.8 | 20.0 |
| Number of WOUDC Sonde Sites | | 20 | 25 | 27 | 30 | 16 | 12 | 16 | 19 |
| Number of Satellite/Sonde Coincidences | | 131 | 197 | 134 | 171 | 72 | 60 | 87 | 128 |

## 5 Additional products under investigation

The current spatial coverage of AIRS+OMI is sufficient to extend the TES ozone record beyond 2010 when TES ceased the global survey mode measurements. The combined AIRS+OMI product can provide a record of tropospheric and total ozone spanning the full Aura satellite time periods (2005 – current). However, the daily global coverage of OMI measurements has been decreasing since 2009 due to the OMI row anomaly (Schenkeveld et al, 2017, Huang et al., 2017; Levelt et al., 2018). Looking to the future and as a way to further increase science return, we have investigated the feasibility of constructing an additional multiple decade long tropospheric ozone profile dataset using a MUSES-based multiple spectral approach that combines the radiance measured by the CrIS and Ozone Mapping Profiler Suite (OMPS) instruments. This additional dataset has the potential to fill the spatial gaps in the joint AIRS+OMI data record since 2012. Both the CrIS and OMPS instruments are on the Suomi National Polar-orbiting Partnership (NPP) satellite, which launched in October 28, 2011. The spectral characteristics of the CrIS instrument (Han et al., 2013; Strow et al., 2013) are similar to the AIRS instrument, and those for OMPS (Flynn et al., 2006, 2014; Kramarova et al., 2014; Pan et al., 2017) are similar to the OMI instrument. Hence, as expected, joint CrIS+OMPS retrievals present characteristics (Figure 7) similar to the joint AIRS+OMI retrievals (Fu et al., 2017).



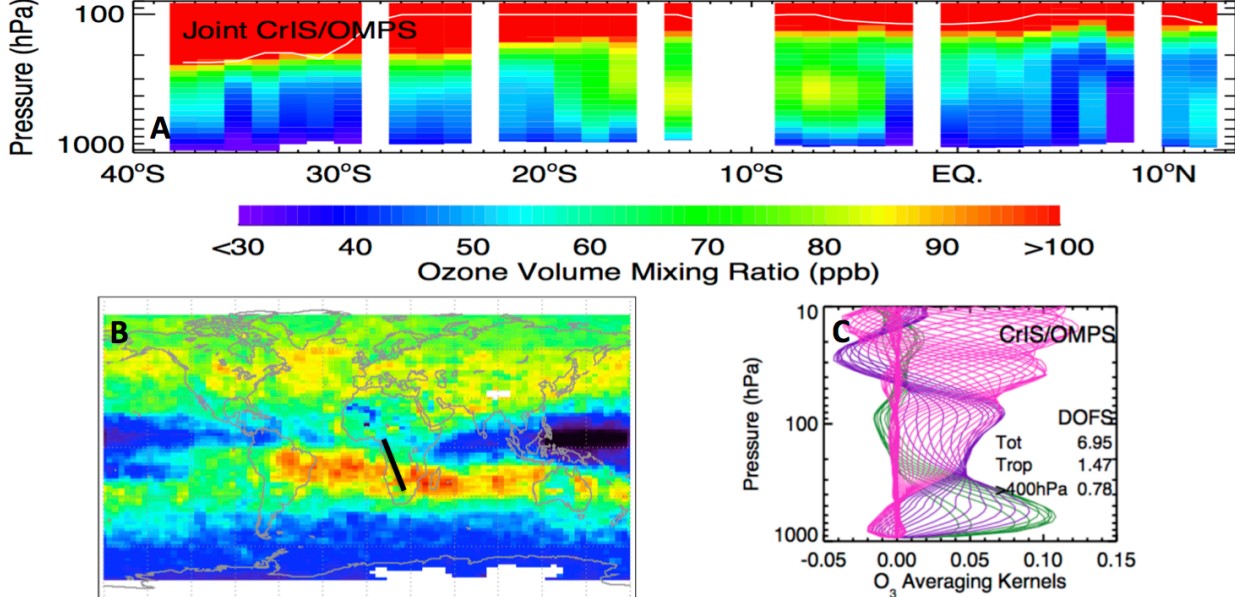

**Figure 7: Joint CrIS+OMPS ozone profile retrievals over Africa on October 21, 2013. The elevated ozone concentrations between 2–20°S are associated with biomass burning. (A) The retrieved ozone concentration profiles along the transect measurements. The white curve indicates the tropopause pressure reported by GEOS-5. (B) TES monthly mean ozone concentration at 510 hPa. The black line indicates the joint CrIS+OMPS measurements location. (C) The averaging kernels of joint CrIS+OMPS measurements.**

## 6 Conclusion

We have demonstrated multispectral retrievals using both AIRS TIR and OMI UV measured radiances for tropospheric $O_3$ profiling. This technique enables the continuation of the TES capability to distinguish between upper and lower tropospheric ozone abundances. The global-scale comparisons between joint AIRS+OMI (version 1) and TES (version 6) $O_3$ profile products across four seasons in the troposphere over global scale, show that these two data products are comparable for a wide variety of geophysical conditions: correlation coefficients are 0.7–0.9 at three pressure levels (316, 510, and 750 hPa), and both the mean (0.8–4.2 ppb) and RMS differences (±4.8–23 ppb) are within the estimated total uncertainties. The joint TIR+UV retrieval provides equivalent vertical sensitivity and error characteristics of TES high spectral measurements, which have a spectral resolution that is ~8–12 times higher than AIRS and OMI measurements though about three times lower SNR. Comparisons of collocated joint AIRS+OMI, TES, and ozonesonde measurements show that both mean and standard deviation of the differences are within the estimated measurement uncertainty of these space sensors. The joint AIRS+OMI ozone products have a high bias of 2–5 ppb similar to TES data (3–6 ppb). Consequently, the similarities of the retrieved concentration, vertical sensitivity and error characteristics between joint AIRS+OMI and TES ozone data, demonstrate that combining the measurements of the existing TIR and UV hyperspectral imaging spectrometers can extend the well-validated NASA EOS high-spectral resolution TES tropospheric ozone profile data products.

Both AIRS and OMI have wide swath widths (AIRS 1,650 km; OMI 2,600 km) across satellites' ground tracks; consequently, the joint AIRS+OMI measurements promise to extend and even improve the number of available observations





by over 100 times that of TES. The product files of the joint AIRS+OMI 2006 ozone global survey retrievals, including a validation report and a reader program are available via the Aura Validation Data Center (AVDC) website (https://avdc.gsfc.nasa.gov/pub/data/satellite/Aura/TES/AIRS_OMI/O3/). The global survey and regional mapping mode of joint AIRS+OMI data from March to June 2016 in support of KORUS-AQ are also available on the same website. These

results have been applied into the post flight data analysis by Miyazaki et al., (2018) that showed great error reductions on the tropospheric ozone analysis, especially in the middle troposphere, through assimilation of joint AIRS+OMI data. Overall comparisons of AIRS+OMI to ozonesondes and aircraft for year 2016 yield similar biases and errors to matching comparisons for year 2006. Using the MUSES algorithm, a TES daily global survey pattern of AIRS+OMI data (2004 to present) is being processed on the facilities within the JPL TES Science Investigator-led Processing (SIP) system to build up a decadal record

of tropospheric ozone products.

It is worth noting that the second set of CrIS and OMPS instruments on board the Joint Polar Satellite System-1 (JPSS-1, also known as NOAA 20) satellite were successfully launched to space on November 18, 2017. The JPSS-2 (also known as NOAA-21) satellite, which is the platform of the third set of CrIS and OMPS instruments, is scheduled to launch in 2022. The NOAA-20/JPSS-1 OMPS Nadir Mapper products' resolution has improved from $50 \times 50$ km$^2$ field of view (FOV)

at nadir to $17 \times 50$ km$^2$, and will further improve to $17 \times 17$ km$^2$ within the next year in the operational NOAA processing (private communication with Dr. Lawrence E. Flynn). The NASA Goddard Space Flight Center (GSFC) level 1 products of JPSS-1 OMPS Nadir Mapper will have a spatial resolution of $10 \times 10$ km$^2$ to help detect sources of sulfur dioxide including volcanoes and coal-burning power plants (press release via spacenews.com by Dr. Glen Jaross). As a result, the joint CrIS+OMPS retrievals, with characteristics similar to AIRS+OMI retrievals but with improved spatial coverage, illustrate the

potentials of extending the tropospheric ozone profile data record to the next decades using the measurements from the Suomi-NPP, JPSS-1, and JPSS-2 satellites. The TROPOMI instrument (Veefkind et al., 2012) on board the sentinel 5 precursor (S5P) satellite successfully deployed into its orbit on October 13 2017 and formed a new satellite constellation with Suomi-NPP, currently 5 minutes apart with the plan of reducing to 3 minutes time difference in the future. The spatial resolution of TROPOMI is an unprecedented $3.5 \times 7.0$ km$^2$ and $7.0 \times 7.0$ km$^2$ in the UV-VIS and shortwave IR (SWIR) spectral bands

accordingly, providing another opportunity of obtaining the high resolution tropospheric ozone and carbon monoxide (Fu et al., 2016) ESDRs via the multispectral retrieval technique that combine CrIS and TROPOMI measurements.

## Acknowledgement

The authors thank Barry L. Lefer, Brendan M. Fisher, Bradley R. Pierce, Brian Drouin, Bryan N. Duncan, Chris D. Barnet, David Crisp, Eric Fetzer, Evan Fishbein, Gordon J. Labow, Helen M. Worden, Irina V. Strickland, Jassim A. Al-saadi, James

H. Crawford, James F. Gleason, Glen Jaross, Jessica L. Neu, Joao Teixeira, Joanna Joiner, Karen Cady-Pereira, Kelly Chance, Krzysztof Wargan, Kuai Le, Lawrence E. Flynn, Larrabee L. Strow, Louisa Emmons, Michael R. Gunson, Monika Kopacz, Nickolay A. Krotkov, Pepijn Veefkind, Pawan K. Bhartia, Richard R. Lay, Richard S. Eckman, Robert J.D. Spurr, Seftor Colin, Scott E. Gluck, Thomas Pagano, Stanley P. Sander, Vivienne H. Payne, and Shanshan Yu for many helpful discussions.





We are grateful to all members of the TES, AIRS, CrIS, OMI, and OMPS instrument, algorithm, validation and science teams for their work on supporting the TES, AIRS, CrIS, OMI, and OMPS missions. We thank Erin Wong and Eugene Y. Chu at JPL for their help on joint AIRS+OMI data production and releasing ozone data files to the NASA AVDC website. We thank Pranjit Saha and Vance R. Haemmerle for their help on the comparisons with the WOUDC ozone data. Support from the

NASA ROSES-2013 Atmospheric Composition: Aura Science Team program (grant number: NNN13D455T) is gratefully acknowledged. Part of the research was carried out at the Jet Propulsion Laboratory, California Institute of Technology, under a contract with the National Aeronautics and Space Administration. K. Miyazaki acknowledges support from JSPS KAKENHI grant numbers 15K05296, 26220101, 26287117, and 18H01285.

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
