# Peer review of "Retrievals of Tropospheric Ozone Profiles from the Synergic Observation of AIRS and OMI: Methodology and Validation"

_Atmospheric Measurement Techniques, 2018_

## Referee Comment (RC1) · Anonymous Referee #2 · 26 Jun 2018

The paper describes an algorithm for retrieving tropospheric ozone concentrations from a synergy of collocated AIRS and OMI spaceborne measurements, named AIRS+OMI. It deals with a challenging methodological problem. The results are presented and discussed only in comparison with TES retrievals and ozonesonde measurements. Overall it provides some evidence of the gain of using the synergetic observation and its performance, but in many occasions it is not precise enough. The paper lacks of sufficient discussion and critical analysis of the results. The improved performances of AIRS+OMI synergetic retrieval are not confronted with single-band IR and UV ozone profile retrievals from AIRS and OMI, which should be the first step of the analysis. The paper is suited for the AMT journal but it requires several major revisions in order to be

publishable.

MAJOR REVISIONS:

1. Comparison of AIRS+OMI with other satellite retrievals: Since the new AIRS+OMI approach is the combination of AIRS and OMI retrievals and such synergism is a challenging task, it is essential to compare the multispectral approach against the single-band methods from AIRS and from OMI. In section 4.1, a comparison of the new AIRS+OMI retrievals is exclusively done with TES, considered as a validated approach. However, it is impossible to know what is the true gain of the multispectral synergy and which information on the vertical profile of ozone is either provided by AIRS or OMI measurements. I strongly recommend comparing the satellite retrievals from AIRS+OMI with AIRS only, OMI only and TES (section 4.1) and also each of them against ozone sondes (in section 4.2).

2. Comparison of AIRS+OMI with ozonesondes: The comparison of satellite retrievals against ozonesondes provided in section 4.2 is too weak to be convincing. The correlation coefficients between satellite and sonde data are not presented in the paper, even if they are essential for evaluating the performance of satellite retrievals. Differences between standard deviations of theses datasets are also an important diagnostic for such evaluations. Moreover, the gain of the multispectral synergism of AIRS+OMS with respect to single-band approaches should be shown in terms of comparisons against ozonesondes. I also recommend also showing the performances of the each of the satellite retrievals against the same ozone sondes and not different datasets, since otherwise we do not know whether the differences come from the chosen datasets or from the performances of the satellite retrievals. This major recommendation is to present a comparison with ozonesonde measurements against, AIRS+OMI, AIRS, OMI and TES, in coincidence for the same sondes (in addition to the full set of ozonesondes coincidences if additional information is provided), and using diagnostics including correlation coefficient and standard deviation comparisons.

3. Additional new multispectral ozone retrievals from CrIS+OMPS: Section 5 of paper presents in an extremely brief and incomplete form a completely new retrieval of ozone from different sounders, corresponding to CrIS+OMPS. I strongly recommend withdrawing this section 5 from the manuscript and all associated conclusions in section 6. The presentation of a new retrieval absolutely needs a thorough comparison against ozonesondes and other satellite retrievals in the same time basis as it is currently done with AIRS+OMI. The title of the paper only refers to AIRS+OMI and not to CrIS+OMPS. It would be much more appropriate to present this new product in a separate paper dedicated to show its performance and its thorough validation.

OTHER GENERAL RECOMMENDATIONS

4. Title of the paper: "synergetic observation of AIRS and OMI" is not clear. I recommend replacing it by " synergism of AIRS and OMI". Also capital letter may only be used at the beginning "Retrieval" and names as AIRS and OMI, but not intermediate words.

5. The good performance of TES ozone retrievals is thoroughly presented in the introduction. I suggest to also mentioning the performance of other tropospheric ozone satellite retrievals different from those from AQUA satellite. It is also stated in the paper that AIRS+OMI retrievals extend the TES record. I think that this association is awkward, since there are many other satellite databases of tropospheric ozone and AIRS+OMI is a different satellite retrieval, independent from TES. I recommend withdrawing the statement "extend the record of TES" of the paper since they are simply two different tropospheric ozone databases and only mentioning that the performance of both AIRS+OMI and TES is similar.

6. Clarity of figures: Many of the figures are difficult to read. It is very difficult to find the labels and quickly see what is shown. I strongly recommend to identify each of the panels of each figure with a letter (a), (b), (c), (d), etc (as the standard way and not A1, A2, B1) outside of the graphs themselves and with clear subtitles also outside of the

graphs (instead of mixing with colors of the plots).

7. Explanations and English language: Many of the panels of the figures are not or very briefly explained (e.g. a priori plots in Fig. 3). This is also the case for biases and RMS differences between satellite retrievals. In many cases, English language is not sufficiently clear or terms appropriately used. Please revise the text in terms of English language and add detail explanations of each of the elements of figures and tables.

8. The comparison of performances of satellite sensors in section 2 should also mention radiometric noise and signal-to-noise ratio and not also spectral resolution.

9. Multispectral retrievals from ozone and carbon monoxide are frequently mentioned together in a single sentence (e.g. conclusions section). These two retrievals are completely independent and different, using different measurements, spectral ranges and configuration of the retrieval techniques. Since the current paper concerns only ozone, I recommend mentioning these retrievals in two separate sentences, first ozone and then carbon monoxide, in a clearer way.

PARTICULAR POINTS:

10. In section 2, it is written that "The spectral resolution of TES is higher than the existing TIR and UV space spectrometers.." Since TES does not measure UV radiances, its spectral resolution against UV sensors is not comparable.

11. In section 2, it is stated: "around local noon time when the atmosphere/land thermal contrast is typically higher than other times of the day. ". What is the evidence for this statement? Is there a reference? This might not always be the case.

12. Page 4 (line 10). The term "healthy" is not objective. Please replace.

13. End of section 2: How is the comparison against ozonesondes is done when several satellite pixels meet the coincidence criteria?

14. Page 4 (lines 23-34): This explanation of the algorithms is confusing and difficult to

read. Please detail separately each part of each of the algorithm. Clearly indicate and explain separately in each case: the radiative transfer codes, the retrieval codes, the a priori profiles and how they are chosen for each individual pixel.

15. Figure 2. Why is the tropopause pressure shown? Is it linked with the choice of a priori profiles for ozone? These aspects should be thoroughly explained.

16. Page 9 : The "species retrieval quality" should be thoroughly detailed in the paper, as a separate paragraph.

17. Section 4.1. The seasonal behavior of the correlation between AIRS/OMI and TES should be explained. Table 3 shows that for all three pressure levels the period of Septembre-Octobre-November coincides with the slight drop of the Pearson correlation coefficient values. Which part of the measurement (AIRS or OMI) is responsible for this? What could be the reason of this behavior? Correlations, biases and RMS between AIRS alone versus TES, and OMI alone versus TES would make it clearer. These comparisons should be added to the discussion.

18. Figure 3: comments of this figure are not precise not clear. They should better indicate the location, the vertical level and the panel to which they are refered.

19. Page 10: For clarity do not interchange the terms "error" and "uncertainty".

20. Page 10 (lines 17-20): This paragraph is difficult to read. Please better explain in a more precise and detailed way.

21. Figure 4 and 5: Please indicate in a legend within the graphs the meaning of the line types.

22. Section 4.2: why number of coincidences between sondes and AIRS+OMI is lower than that for TES? Please clarify in detail the spatial coverage, pixels sizes, percentage of quality assured retrievals with respect to total pixels available, for all retrievals: TES, OMI, AIRS and AIRS+OMI.

23. Conclusions in section 6: "We have demonstrated multispectral retrievals ..." is awkward. Please change by "We have SHOWN or PRESENTED ..."

24. Conclusions in section 6. The statement "The product files of the joint AIRS+OMI 2006 ozone global survey retrievals, including a validation report and a reader program are available via the Aura Validation Data Center (AVDC) website (https://avdc.gsfc.nasa.gov/pub/data/satellite/Aura/TES/AIRS_OMI/O3/). " should be corrected. This webpage contains the error message 'This file you are trying to access was not found on the server.'

25. Page 16 (also before): the "global survey" and "regional mapping" modes are not clear. What is the meaning of these modes? Are they "gridding" or "sampling" of the retrievals? This should be better explain and detail.

26. Page 16 (line 8): the term "a TES daily global survey pattern of AIRS+OMI data" is not clear. What does it mean? If it is gridding or sample, I strongly recommend simply specifying the resolution of the AIRS+OMI data and then mentioning that is the same resolution as TES.

---

## Referee Comment (RC2) · Anonymous Referee #1 · 23 Jul 2018

I find the manuscript well written and of good scientific significance. I would like to make few suggestions that in my opinion could strengthen the content of the manuscript. These pertain essentially section 3 which could greatly benefit from a better elucidation of some of the key aspects of a retrieval method development, such as its resilience to clouds, global yield, dependence from the a priori, sources of uncertainty. Please read below.

1) Equation 3). The author did not expand on the yWˆTW term, specifically how it is built and the role it plays in the retrieval convergence.

2) Page 6, point 5 and 6. Are near real time model data used in the scheme or an

offline climatology built on these sources? What does "estimated from MOZART" really means?

3) Equation 4, page 7. Is the epsilon term simply the instrument noise? How is the cross-state term exactly built? Hence, in equation (6), what are the terms entering the S_epsilon matrix? How are S_cs and A_sc built? Does this retrieval scheme take into account for the radiative transfer model uncertainty as well in the measurement error covariance?

4) Page 9. Can the author describe the "species retrieval quality"? How is the effective cloud optical depth computed and what's its uncertainty? Is this uncertainty taken into account in the retrieval scheme? How is the cloud fraction computed? Is this an independent computation and if not, what's its uncertainty and is it factored in the retrieval scheme anywhere? For example, is its error covariance gaussian?

5) Page 9. Only daytime scenes are included. Does this mean that this retrieval scheme only applies to day time scenes in general? Can the authors specify this aspect? Also, is the retrieval scheme only applicable to cases with effective OMI cloud fraction less then 30%? If so, what is the overall global yield of the proposed retrieval scheme? Figure 3 is somehow ambiguous. Restricting the applicability of the retrieval to scenes that are less than 30% cloudy does not seem to correspond to what is displayed in figure 3, where the retrieval acceptance yield seems to be in reality 100% globally. What day was used for this figure exactly? Are those multiple days overlapped?

6) Equation 7. The term GS_erGˆT was not included in the earlier equations.

7) Typo. Line 20. AIRS+OMI, not AIR+OMI.
* * *

---

## Author Comment (AC2) · 24 Aug 2018

The comment was uploaded in the form of a supplement. Thanks.

Please also note the supplement to this comment:
https://www.atmos-meas-tech-discuss.net/amt-2018-138/amt-2018-138-AC2-supplement.zip

---

## Author Response (AR1)

We made the following changes to the text as suggested by reviewer #1 (review comments are in *italic*; all the changes are in blue). We thank the reviewer very much for the useful comments.

**Reviewer #1's suggestions:**

*I find the manuscript well written and of good scientific significance. I would like to make few suggestions that in my opinion could strengthen the content of the manuscript. These pertain essentially section 3 which could greatly benefit from a better elucidation of some of the key aspects of a retrieval method development, such as its resilience to clouds, global yield, dependence from the a priori, sources of uncertainty. Please read below.*

**Re:** We added some discussions in Page 6 lines 20 to 25, Page 7 line 11.

"The joint AIRS+OMI retrievals start with the list of the fitting parameters, a priori values, and a priori variance shown in Table 2. In addition to the initial guess for the trace gas concentration ($O_3$, $H_2O$, and $CO_2$), the initial guess for auxiliary parameters used in the simulation of AIRS radiances (including temperature profile, surface temperature and emissivity, cloud extinction and cloud top pressure) are also retrieved from AIRS radiances in order to take into account their spectral signatures in the $O_3$ spectral regions. The joint AIRS+OMI algorithm incorporated a suite of treatments in order to optimize the spatial resolution, retrieval stability, data throughput, and consistency to TES data products (version 6): (1) When the clouds travel across its field of view, a space sensor for atmospheric composition measurements often faces to the challenge of obtaining high precision and accuracy measurements of the trace gas vertical distribution due to the interference among retrieval parameters. MUSES algorithm uses single-footprint AIRS level 1B radiances in the retrievals (Irion et al., 2018), which leads to a footprint nine times smaller in area than the AIRS version 6 operational algorithm (Susskind et al., 2003 and 2014), partially mitigating the chance of the impacts of cloud interference on the trace gas retrievals; (2) global infrared land surface emissivity database from the University of Wisconsin-Madison (UOW-M) (Seemann et al., 2007), which improves clear land throughput by 4.5%; (3) an initial guess refinement step of cloud fraction prior to the step of joint AIRS+OMI ozone retrievals; (4) a priori constraint vector and matrix identical to the TES version 6 operational algorithm to obtain error estimates consistent with TES data products; (5) an updated a priori and initial guess information of atmospheric temperature profiles taken from the near real time Goddard Earth Observing System Model, Version 5 (GEOS-5) (Rienecker et al., 2008) model data for AIRS TIR temperature profile retrievals; (6) updated a priori ozone built from the Model for OZone and Related chemical Tracers (MOZART)-4 (Emmons et al., 2010) as offline climatology; (7) HIgh-resolution TRANsmission (HITRAN) 2012 (Rothman et al., 2013) spectroscopic parameters and a priori information of water vapor, the primary interfering species in TIR ozone measurements jointly retrieved with ozone; and (8) the target scenes with retrieved cloud fraction less than 30% within AIRS+OMI field of view are labeled as good quality, in order to minimize the impacts of cloud interference on ozone data quality. The throughput of AIRS+OMI data processing over the globe is about 30%."

*1) Equation 3). The author did not expand on the $\gamma_i \mathbf{W}^\mathrm{T}\mathbf{W}$ term, specifically how it is built and the role it plays in the retrieval convergence.*

**Re:** We added some discussions in Page 5 line 24-26 and Page 6 line 1-2.

"The computation of the $\gamma_i$ value and $\mathbf{W}$ follow the sections 5.5 and 6.3 of Moré (1977), utilizing the fitting residual and $\mathbf{K}$ from the space instruments as input parameters. The $\gamma_i \mathbf{W}^\mathrm{T}\mathbf{W}$ term, the

core of the trust-region LM optimization algorithm, plays the crucial role in balance the convergence speed and robustness. Under large $\gamma_i$ , the step size computation is similar to a steepest descent algorithm, which has a lower convergence rate, and under low $\gamma_i$ , the step computation is towards a Gauss-Newton approach."

*2) Page 6, point 5 and 6. Are near real time model data used in the scheme or an offline climatology built on these sources? What does "estimated from MOZART" really means?*
**Re:** We added some discussions in Page 7 line 4-7.
"(5) an updated a priori and initial guess information of atmospheric temperature profiles taken from the near real time Goddard Earth Observing System Model, Version 5 (GEOS-5) (Rienecker et al., 2008) model data for AIRS TIR temperature profile retrievals; (6) updated a priori ozone built from the Model for OZone and Related chemical Tracers (MOZART)-4 (Emmons et al., 2010) as offline climatology;".

*3) Equation 4, page 7. Is the epsilon term simply the instrument noise? How is the cross-state term exactly built? Hence, in equation (6), what are the terms entering the S_epsilon matrix? How are S_cs and A_sc built? Does this retrieval scheme take into account for the radiative transfer model uncertainty as well in the measurement error covariance?*
**Re:** For equation 4, page 7, we revised Page 8 lines 5 to 8.
"$\hat{\mathbf{x}} = \mathbf{x}_a + \mathbf{A}[\mathbf{x}_{true} - \mathbf{x}_a] + \mathbf{G}\varepsilon + \delta_{cs},$ (4)

[revised manuscript text omitted]

*6) Equation 7. The term $GS_{\varepsilon}G^T$ was not included in the earlier equations.*
**Re:** The $GS_{\varepsilon}G^T$ term was included in the earlier equation 6, prior to equation 7.

*7) Typo. Line 20. AIRS+OMI, not AIR+OMI.*
**Re:** We changed to "AIRS+OMI".

We made the following changes to the text as suggested by reviewer #2 (review comments are in *italic*; all the changes are in blue). We thank the reviewer very much for the useful comments.

**Reviewer #1's suggestions:**
*The paper describes an algorithm for retrieving tropospheric ozone concentrations from a synergy of collocated AIRS and OMI spaceborne measurements, named AIRS+OMI. It deals with a challenging methodological problem. The results are presented and discussed only in comparison with TES retrievals and ozonesonde measurements. Overall it provides some evidence of the gain of using the synergetic observation and its performance, but in many occasions, it is not precise enough. The paper lacks of sufficient discussion and critical analysis of the results. The improved performances of AIRS+OMI synergetic retrieval are not confronted with single-band IR and UV ozone profile retrievals from AIRS and OMI, which should be the first step of the analysis. The paper is suited for the AMT journal but it requires several major revisions in order to be publishable.*

Re: We revised the sections 2-5, figures and tables, in order to provide more details on the characteristics/value of other data sets, the community recognition on the value of multiple spectral approach, and the incorporation of ozonesonde-single space sensors comparisons.

*MAJOR REVISIONS:*
*1. Comparison of AIRS+OMI with other satellite retrievals: Since the new AIRS+OMI approach is the combination of AIRS and OMI retrievals and such synergism is a challenging task, it is essential to compare the multispectral approach against the single-band methods from AIRS and from OMI. In section 4.1, a comparison of the new AIRS+OMI retrievals is exclusively done with TES, considered as a validated approach. However, it is impossible to know what is the true gain of the multispectral synergy and which information on the vertical profile of ozone is either provided by AIRS or OMI measurements. I strongly recommend comparing the satellite retrievals from AIRS+OMI with AIRS only, OMI only and TES (section 4.1) and also each of them against ozone sondes (in section 4.2).*

**Re:** We made revisions sections 2, 3, and 4, in order to provide more details on the characteristics/value of other data sets, the community recognition on the value of multiple spectral approach, and the incorporation of ozonesonde-single space sensors comparisons. We added in the comparisons of AIRS-sonde and OMI-sonde comparisons, for both with and without satellite observation operators applied to the ozonesonde profiles.

Please note that, the scientific community/publics have been aware of the values of multiple spectral approach for the tropospheric ozone profiling, via the existing publication - both theoretical study (Landgraf *et al.* J. Geophys. Res. 2007; Worden *et al.* Geophys. Res. Lett. 2007b) and using actual measurements including joint TES+OMI (Fu *et al.* ACP 2013) and joint IASI+GOME2 (Cuesta *et al.* ACP 2013, 2018) from space. And in section 3.2, this work has conducted the quantitative comparisons among all three data sets (joint AIRS+OMI, AIRS alone, OMI alone) via evaluation of both the vertical resolution/sensitivity and retrieved ozone profiles. Intensive data processing for AIRS alone and OMI alone measurements are not the goal of this work, since the AIRS alone and OMI alone showed significant lower sensitivity than TES and joint AIRS+OMI, lacking of the desirable characteristics identified by the data user community of tropospheric ozone profiles. In addition, we extended satellite-ozonesonde comparisons in section 4.2 by adding the comparisons single band retrievals (AIRS alone; OMI alone) to the ozonesonde measurements. The joint AIRS+OMI retrievals show reduced biases, in comparison to the single

band data sets. Detailed information is provided via the updated text in section 4.2; updated Figure 6 and Table 4; and the Table 5 that is added in this section.

*2. Comparison of AIRS+OMI with ozonesondes: The comparison of satellite retrievals against ozonesondes provided in section 4.2 is too weak to be convincing. The correlation coefficients between satellite and sonde data are not presented in the paper, even if they are essential for evaluating the performance of satellite retrievals. Differences between standard deviations of theses datasets are also an important diagnostic for such evaluations. Moreover, the gain of the multispectral synergism of AIRS+OMS with respect to single-band approaches should be shown in terms of comparisons against ozonesondes. I also recommend also showing the performances of the each of the satellite retrievals against the same ozone sondes and not different datasets, since otherwise we do not know whether the differences come from the chosen datasets or from the performances of the satellite retrievals. This major recommendation is to present a comparison with ozonesonde measurements against, AIRS+OMI, AIRS, OMI and TES, in coincidence for the same sondes (in addition to the full set of ozonesondes coincidences if additional information is provided), and using diagnostics including correlation coefficient and standard deviation comparisons.*

**Re:** As stated in the response of suggestion #1, we extended satellite-ozonesonde comparisons in section 4.2 by adding the comparisons single band retrievals (AIRS alone; OMI alone) to the sonde measurements. The joint AIRS+OMI retrievals show reduced biases, in comparison to the single band data sets.

*3. Additional new multispectral ozone retrievals from CrIS+OMPS: Section 5 of paper presents in an extremely brief and incomplete form a completely new retrieval of ozone from different sounders, corresponding to CrIS+OMPS. I strongly recommend withdrawing this section 5 from the manuscript and all associated conclusions in section 6. The presentation of a new retrieval absolutely needs a thorough comparison against ozonesondes and other satellite retrievals in the same time basis as it is currently done with AIRS+OMI. The title of the paper only refers to AIRS+OMI and not to CrIS+OMPS. It would be much more appropriate to present this new product in a separate paper dedicated to show its performance and its thorough validation.*

**Re:** We removed the section 5. Please note that the retrieval algorithm used for processing CrIS+OMPS is identical to that used for AIRS+OMI. In order to info the community on the future application of joint AIRS+OMI retrieval algorithm, we reported our prototype retrievals of CrIS+OMPS. This is for illustrating the future application of this work. Hence, it is not necessary to include CrIS+OMPS in the title nor the intensive data processing of joint CrIS+OMPS data. The detailed characterization and validation of CrIS+OMPS data products will be presented in a separate paper/follow on work.

*OTHER GENERAL RECOMMENDATIONS*
*4. Title of the paper: "synergetic observation of AIRS and OMI" is not clear. I recommend replacing it by "synergism of AIRS and OMI". Also capital letter may only be used at the beginning "Retrieval" and names as AIRS and OMI, but not intermediate words.*

**Re:** Revised the title to "Retrievals of tropospheric ozone profiles from the synergism of AIRS and OMI: methodology and validation".

*5. The good performance of TES ozone retrievals is thoroughly presented in the introduction. I suggest to also mentioning the performance of other tropospheric ozone satellite retrievals different from those from AQUA satellite. It is also stated in the paper that AIRS+OMI retrievals extend the TES record. I think that this association is awkward, since there are many other satellite databases of tropospheric ozone and AIRS+OMI is a different satellite retrieval, independent from TES. I recommend withdrawing the statement "extend the record of TES" of the paper since they are simply two different tropospheric ozone databases and only mentioning that the performance of both AIRS+OMI and TES is similar.*

**Re:** Page 2 line 6 to 12, we discussed the ozone data products from OMI, which differ from those from Aqua satellite.

"Extending the record of TES" is an ongoing task endorsed by the user community of tropospheric ozone profile data products and funded by NASA program office in order to meet the scientific and programmatic needs of continuing the TES data record. Since the similarity of the performances between joint AIRS+OMI and TES, joint AIRS+OMI ozone data could be used to extend the record of TES.

*6. Clarity of figures: Many of the figures are difficult to read. It is very difficult to find the labels and quickly see what is shown. I strongly recommend to identify each of the panels of each figure with a letter (a), (b), (c), (d), etc (as the standard way and not A1, A2, B1) outside of the graphs themselves and with clear subtitles also outside of the graphs (instead of mixing with colors of the plots).*

**Re:** Combining two reviewers' comments and taken the following facts into account, we revised the Figures 4 and 5 as well as supplement Figures S12–S33. We revised figure 3 and supplement Figures S1–S11 via identifying each of the panels of the graphs with subtitles outside of the graphs. Prior to submit this manuscript to AMTD in April 2018, we had two technical writer editors who have bachelor degree on English and working in our institute for document services, reviewed this manuscript to have quality ensured. And we also had all coauthors contribute to help on the editing the entire manuscript.

*7. Explanations and English language: Many of the panels of the figures are not or very briefly explained (e.g. a priori plots in Fig. 3). This is also the case for biases and RMS differences between satellite retrievals. In many cases, English language is not sufficiently clear or terms appropriately used. Please revise the text in terms of English language and add detail explanations of each of the elements of figures and tables.*

**Re:** Responses available in comment #6. We updated all figures in the main text and supplement figures.

*8. The comparison of performances of satellite sensors in section 2 should also mention radiometric noise and signal-to-noise ratio and not also spectral resolution.*

**Re:** We added in the discussion in page 3 lines 25 to 27.

"Taking the spectral coverage, spectral resolution, and noise performance into account, the vertical sensitivity of TES and other satellite sensors (AIRS alone, OMI alone) is quantified in section 3.2. It shows that TES has the sensitivity to distinguish between the upper and lower tropospheric $O_3$."

*9. Multispectral retrievals from ozone and carbon monoxide are frequently mentioned together in a single sentence (e.g. conclusions section). These two retrievals are completely independent and*

*different, using different measurements, spectral ranges and configuration of the retrieval techniques. Since the current paper concerns only ozone, I recommend mentioning these retrievals in two separate sentences, first ozone and then carbon monoxide, in a clearer way.*

**Re:** We deleted the carbon monoxide in the conclusion section.

"The spatial resolution of TROPOMI is an unprecedented $3.5 \times 7.0$ km$^2$ and $7.0 \times 7.0$ km$^2$ in the UV-VIS and shortwave IR (SWIR) spectral bands accordingly, providing another opportunity of obtaining the high-resolution tropospheric ozone ESDR via the multispectral retrieval technique that combine CrIS and TROPOMI measurements."

*PARTICULAR POINTS:*

*10. In section 2, it is written that "The spectral resolution of TES is higher than the existing TIR and UV space spectrometers.." Since TES does not measure UV radiances, its spectral resolution against UV sensors is not comparable.*

**Re:** We revised to "The spectral resolution of TES (resolving power (RP) 10,500) is significantly higher than the existing TIR including AIRS (RP: 1,200), CrIS (RP: 816), IASI (RP: 5,250).".

*11. In section 2, it is stated: "around local noon time when the atmosphere/land thermal contrast is typically higher than other times of the day. ". What is the evidence for this statement? Is there a reference? This might not always be the case.*

**Re:** This statement illustrates a known fact, which have been written in many text books and radiative transfer modeling and remote sensing technologies.

*12. Page 4 (line 10). The term "healthy" is not objective. Please replace.*

**Re:** We change "the OMI "healthy" off-nadir pixels" to "the quality-assured OMI off-nadir pixels".

*13. End of section 2: How is the comparison against ozonesondes is done when several satellite pixels meet the coincidence criteria?*

**Re:** We compute the differences for each satellite-sonde pair, then do the analysis, i.e., we would use all the available satellite results as long as their quality-assured.

*14. Page 4 (lines 23-34): This explanation of the algorithms is confusing and difficult to read. Please detail separately each part of each of the algorithm. Clearly indicate and explain separately in each case: the radiative transfer codes, the retrieval codes, the a priori profiles and how they are chosen for each individual pixel.*

**Re:** Combining two reviewers' comment and taken the following facts into account, we would keep the lines 23-34 without having further revision. (1) Prior to submit this manuscript to AMTD in April 2018, we had two technical writer editors who have bachelor degree on English and working in our institute for document services, reviewed this manuscript to have quality ensured. And we also had all coauthors contribute to help on the entire manuscript. (2) The references cited in this paragraph provide the detailed information on the heritage of the radiative transfer codes, the retrieval codes, the a priori profiles etc. Hence, this paper does not repeat the information published in the previous works.

*15. Figure 2. Why is the tropopause pressure shown? Is it linked with the choice of a priori profiles for ozone? These aspects should be thoroughly explained.*

**Re:** The values of tropopause pressure are provided in the joint AIRS+OMI ozone data product files, as shown in Figure 2. Conventionally, the tropopause pressure value needed for a few scientific applications, e.g., studying the processes/impacts of stratosphere-troposphere exchange on the tropospheric ozone abundances.

*16. Page 9: The "species retrieval quality" should be thoroughly detailed in the paper, as a separate paragraph.*
**Re:** We revised the paragraphs thoroughly shown below. The updated text from line 10, Page 10 to line 5, Page 11.

"Joint AIRS+OMI ozone retrievals apply to only daytime scenes, since OMI measurements depend on the sunlight, though the MUSES algorithm processes both day time and night time TIR space measurements. The "species retrieval quality" flag of joint AIRS+OMI data, – a master quality flag available in the level 2 product files, was determined by evaluating a suite of retrieval characteristics including the spectral fitting residuals, cloud fraction within field of view (when effective cloud fraction in OMI > 30%), and the lapse rate of tropospheric ozone vertical distribution. The retrieval scheme processes the AIRS+OMI measurements over all sky conditions, though only the scenes of the cloud fraction within field of view less than 30% were flagged as good quality. The retrieval acceptance rate of joint AIRS+OMI ozone in 2006 is about 30%.

Both TES and joint AIRS+OMI 2006 ozone profile data were screened prior to the comparison using (1) the ''species retrieval quality''; (2) the retrieved cloud effective TIR optical depth (removed when OD > 2.0); (3) solar zenith angle (SZA; excluded when SZA > 80°, i.e., day time only). We excluded profiles with thick clouds in the field of view because these obscures the infrared emission from the lower troposphere, which greatly reduces the satellite sensitivity of both TIR and UV radiances. For cloud treatment, we adopt the approach used in the joint TES+OMI retrieval algorithm (Fu et al., 2013) by adding in an initial guess refinement step for retrieving the cloud fraction within OMI field of view, prior to joint AIRS+OMI ozone retrievals. The impacts of cloud and surface properties have been taken account into the retrievals, since the MUSES algorithm simultaneously retrieve both the trace gases profiles and the cloud/surface parameters. The retrieved values and estimated errors of the cloud effective TIR optical depth and cloud height, UV cloud fraction within the field of view and cloud top height are provided in the joint AIRS+OMI data product files."

*17. Section 4.1. The seasonal behavior of the correlation between AIRS/OMI and TES should be explained. Table 3 shows that for all three pressure levels the period of Septembre-Octobre-November coincides with the slight drop of the Pearson correlation coefficient values. Which part of the measurement (AIRS or OMI) is responsible for this? What could be the reason of this behavior? Correlations, biases and RMS between AIRS alone versus TES, and OMI alone versus TES would make it clearer. These comparisons should be added to the discussion.*
**Re:** We added the following discussion in Page 11 lines 10 -17.
"The period of September-October-November coincides show the slight drop of the Pearson correlation coefficient values. For September 2006 data, the different spatial/temporal sampling between TES and joint AIRS+OMI data is the reason for the slight drop. In September 2006, TES and joint AIRS+OMI data delivers nine and fifteen global surveys accordingly (bottom row of Table 3). TES did not deliver measurements from September 1 to 9. For supporting the TEXAQS II flight campaign, TES delivered additional special observations by reducing the number of

global surveys in the end of September. For October and November 2006 data, the slight drop of the correlation coefficients might relate to the slight difference of measurement sensitivity between TES and joint AIRS+OMI, as shown in supplement figures S20 and S21."

*18. Figure 3: comments of this figure are not precise not clear. They should better indicate the location, the vertical level and the panel to which they are refered.*
**Re:** Responses available in comment #6.

*19. Page 10: For clarity do not interchange the terms "error" and "uncertainty".*
**Re:** We updated the text in order to keep the terms being consistent.

*20. Page 10 (lines 17-20): This paragraph is difficult to read. Please better explain in a more precise and detailed way.*
**Re:** We revised that paragraph, now it is in Page 12 line 4 to 12.
"The characteristics of the joint AIRS+OMI retrievals, in terms of vertical sensitivity and estimated error characteristics, are similar to those of TES data. The DOFS, which quantify the vertical sensitivity of global tropospheric ozone retrievals, show distributions similar to TES data (Figs. 4 panels A2 and B2 for August 2006). Supplement Figures S12–S22 presented the DOFS for the remaining months of 2006. Both the estimated observation and total errors of joint AIRS+OMI retrievals (black curves of Fig. 5) show peaks and widths equivalent to that of TES data products (green curves of Fig. 5) across troposphere over the globe. Supplement Figures S23–S33 presented the estimated errors for the remaining months of 2006. The peak of the estimated observation errors, which are the sum of second and third terms in Eq, (6), reside in the range of 6–8% (or ~3 ppb) for the joint AIRS+OMI retrievals – equivalent to the observation error of 5–7% (or ~2–3 ppb) from TES data across the troposphere. Finally, the joint AIRS+OMI retrievals have total errors within 3% agreement over the globe - equivalent to TES data."

*21. Figure 4 and 5: Please indicate in a legend within the graphs the meaning of the line types.*
**Re:** We updated the Figures 4 and 5, as well as supplement Figures S12–S33.

*22. Section 4.2: why number of coincidences between sondes and AIRS+OMI is lower than that for TES? Please clarify in detail the spatial coverage, pixels sizes, percentage of quality assured retrievals with respect to total pixels available, for all retrievals: TES, OMI, AIRS and AIRS+OMI.*
**Re:** We deleted "" in line 16, Page 4 and revised line 15 to 20, Page 4 for clarification.
"To examine the performances of remote sensing measurements, we applied the following coincidence criteria to determine sonde-AIRS+OMI: (1) mean cloud optical depth < 2.0, (2) cloud fraction within OMI field of view < 30%, (3) both satellite ground pixel-sonde distances < 300 km, (4) solar zenith angle < 80°,  and (5) daytime measurements with a time difference < 4 hour. In order to determine the sonde-TES pairs, we applied the criteria (1), (3), (4) and (5), and excluded the criteria (2) since the TES retrievals do not use information from OMI measurements. As a result, for the 2006 timeframe, we obtained 424 sonde-AIRS+OMI triads and 556 sonde-TES measurement pairs."

*23. Conclusions in section 6: "We have demonstrated multispectral retrievals :::" is awkward. Please change by "We have SHOWN or PRESENTED :::"*
**Re:** We changed "demonstrated" to "shown" in Page 17 line 6.

*24. Conclusions in section 6. The statement "The product files of the joint AIRS+OMI 2006 ozone global survey retrievals, including a validation report and a reader program are available via the Aura Validation Data Center (AVDC) website (https://avdc.gsfc.nasa.gov/pub/data/satellite/Aura/TES/AIRS_OMI/O3/). " should be corrected. This webpage contains the error message 'This file you are trying to access was not found on the server.'*
**Re:** Some updates on the directory structure of ADVC website have been made recently, after this manuscript submitted to AMTD. As a result, the old website was no longer available. We updated the link in Page 18 line 3 of revised manuscript, shown below.
"The product files of the joint AIRS+OMI 2006 ozone global survey retrievals, including a validation report and a reader program are available via the Aura Validation Data Center (AVDC) website (https://avdc.gsfc.nasa.gov/pub/data/satellite/Aura/TES/AIRS_OMI-version0.1Beta/)."

*25. Page 16 (also before): the "global survey" and "regional mapping" modes are not clear. What is the meaning of these modes? Are they "gridding" or "sampling" of the retrievals? This should be better explain and detail.*
**Re:** The description of "global survey" and "regional mapping" modes are available in Page 9 line 18 to 22, with revision shown below.
"Further evaluation of the joint AIRS+OMI $O_3$ retrievals are shown in two modes: Global Survey (GS) and REgional mapping (RE). The GS mode provides profile data at nadir position along the satellite ground track, i.e., a temporal/spatial sampling identical to TES GS, while RE mode processes all available AIRS+OMI measurements over a region, specifically in this case we have considered the Korean peninsula during the 2016 KORUS-AQ campaign (Miyazaki et al., 2018)."

*26. Page 16 (line 8): the term "a TES daily global survey pattern of AIRS+OMI data" is not clear. What does it mean? If it is gridding or sample, I strongly recommend simply specifying the resolution of the AIRS+OMI data and then mentioning that is the same resolution as TES.*
**Re:** The global survey data is not gridded. We revised the referred text in Page 18 line 2 to 3, shown below.
"Using the MUSES algorithm, the AIRS+OMI global survey mode data (2004 to present) with a footprint size about 15 by 24 km is being processed on the facilities within the JPL TES Science Investigator-led Processing (SIP) system to build up a decadal record of tropospheric ozone products."